# Smad4 controls signaling robustness and morphogenesis by differentially contributing to the Nodal and BMP pathways

Luca Guglielmi[1], Claire Heliot[1], Sunil Kumar [2], Yuriy Alexandrov[2], Ilaria Gori[1], Foteini Papaleonidopoulou [1], Christopher Barrington [3], Philip East [3], Andrew D. Economou[1], Paul M. W. French[4], James McGinty[4] & Caroline S. Hill [1✉]

The transcriptional effector SMAD4 is a core component of the TGF-β family signaling pathways. However, its role in vertebrate embryo development remains unresolved. To address this, we deleted Smad4 in zebrafish and investigated the consequences of this on signaling by the TGF-β family morphogens, BMPs and Nodal. We demonstrate that in the absence of Smad4, dorsal/ventral embryo patterning is disrupted due to the loss of BMP signaling. However, unexpectedly, Nodal signaling is maintained, but lacks robustness. This Smad4-independent Nodal signaling is sufficient for mesoderm specification, but not for optimal endoderm specification. Furthermore, using Optical Projection Tomography in combination with 3D embryo morphometry, we have generated a BMP morphospace and demonstrate that Smad4 mutants are morphologically indistinguishable from embryos in which BMP signaling has been genetically/pharmacologically perturbed. Smad4 is thus differentially required for signaling by different TGF-β family ligands, which has implications for diseases where Smad4 is mutated or deleted.

---

[1] Developmental Signalling Laboratory, The Francis Crick Institute, London NW1 1AT, UK. [2] Advanced Light Microscopy, The Francis Crick Institute, London NW1 1AT, UK. [3] Bioinformatics and Biostatistics Facility, The Francis Crick Institute, London NW1 1AT, UK. [4] Department of Physics, Imperial College London, SW7 2AZ London, UK. ✉email: caroline.hill@crick.ac.uk

Morphogens are extracellular signaling molecules that act locally or at a distance to instruct and coordinate cell fate decisions across space and time[1,2]. Specific features of the signal, such as concentration and duration, are interpreted by receiving cells via highly regulated cellular responses[1]. These are typically mediated by intracellular effectors, transcription factors, and cofactors[3]. The transcriptional effector SMAD4 mediates signaling responses to the transforming growth factor β (TGF-β) family of morphogens in a variety of biological contexts, ranging from embryo development to adult tissue homeostasis[4–6]. Consistent with this central role, loss of SMAD4 is associated with human disease, most commonly cancer, where it acts as a prominent tumor suppressor[7–9].

SMAD4 is described as the "common SMAD", given its involvement in signal transduction in both arms of the TGF-β family signaling pathways; the SMAD1/5 arm, which is canonically activated by BMPs and some GDF ligands, and the SMAD2/3 arm, activated by NODAL, Activins, TGF-β, and other GDF family members[10,11]. Upon ligand binding, receptor-regulated SMADs (R-SMADs) SMAD1/5 and SMAD2/3 are phosphorylated by activated type I receptors and form complexes with SMAD4 that accumulate in the nucleus to regulate target gene expression[10,11]. The SMAD complexes have low affinity for DNA and cooperate with other transcription factors to regulate transcription[12]. Phosphorylated SMAD1/5 (pSMAD1/5) and pSMAD3 bind DNA directly with SMAD4 at specific SMAD-binding elements[12]. pSMAD2–SMAD4 complexes, in contrast, do not bind DNA directly but are recruited by other transcription factors, the first characterized being the forkhead transcription factor, FOXH1 (formerly called Fast1)[13,14]. Whilst the in vitro data has established a central role for SMAD4 in these signal transduction pathways, the functional role of SMAD4 in vivo has not been fully resolved.

The signaling requirement for SMAD4 has mostly been explored in developmental systems in the context of well-characterized responses to the BMP and NODAL morphogens, which pattern embryos into the three discrete germ layers during gastrulation. NODAL signaling is required for mesoderm and endoderm specification and left–right asymmetry[15–18], whereas BMP signaling patterns the ectoderm and is essential for dorsal/ventral (D/V) patterning[19,20]. The necessity of SMAD4 for NODAL and BMP signaling has so far been inconclusive, with studies reporting SMAD4-independent responses for each arm of the pathway. For example, in Drosophila, the SMAD4 homolog Medea is essential for D/V patterning of the embryo, as Medea mutants display severe axis ventralization, a phenotype similar to embryos mutant for the BMP homolog decapentaplegic (Dpp)[21,22]. While this suggested a general signaling requirement for Medea in Dpp signaling, analysis of Medea mutant clones revealed that in the developing wing disc, Dpp signaling functions independently of Medea close to the Dpp source, while at increasing distances, Medea is essential for Dpp-induced target gene expression[23]. Similarly, females bearing Medea-null clones in the germline can produce fertilized eggs, a process dependent on Dpp and the SMAD1/5 homolog, Mad[23].

In the mouse, global loss of SMAD4 is embryonically lethal, with the embryos dying at the onset of gastrulation[24]. However, conditional loss of SMAD4 in the epiblast generated embryos that gastrulated but displayed mixed outcomes in terms of patterning. Specifically, these mutants failed to form streak derivatives induced by NODAL, like the node, notochord, prechordal plate, and definitive endoderm. However, they developed an allantois and a mis-patterned heart, which are under the control of BMP signaling[4]. This phenotype is reminiscent of a complete loss of NODAL signaling in the epiblast[25,26]. Nevertheless, interpretation of these data is complicated by the fact that in the mouse

embryo BMPs and NODAL regulate each other, since epiblast-derived NODAL can induce BMP4 synthesis in the extraembryonic ectoderm which in turn promotes NODAL expression in the epiblast[17,27–29]. Therefore, while evidence in the mouse clearly demonstrates the existence of SMAD4-independent responses, it is not resolved whether they occur downstream of BMPs, NODAL or both.

To address this question, we have exploited the zebrafish embryo where patterning by BMPs and Nodal is well described, and these ligands are induced independently of each other[30]. The Nodal ligands Ndr1/2 are secreted by the extraembryonic yolk syncytial layer (YSL) to induce their own expression in the overlaying blastoderm margin[18,31–33]. This results in a signaling gradient which extends about five cell tiers[33,34]. Both the size and amplitude of this signaling domain are finely tuned by the Nodal antagonists, Lft1/2, also expressed under the control of Nodal in a negative feedback mechanism[33,35]. Expression of the zygotic BMP ligands Bmp2b/4/7a is primarily induced by maternal Gdf6a, which is a BMP subfamily member that signals through the type I BMP receptor Alk8 and Smad5[36–40]. BMP signaling levels are initially uniform within the blastula, but are progressively cleared from the dorsal side of the embryo by dorsal expression of the BMP antagonist Chordin[41], through a proposed source–sink mechanism[42,43].

Here, we generate a Smad4 mutant zebrafish line and systematically explore the consequences of Smad4 loss downstream of Nodal and BMP signaling. Contrary to the classical view, we find that in maternal-zygotic (MZ) smad4a mutants, Nodal signaling at mid gastrulation is broadly maintained at levels similar to wild type (WT). We show that the initial Smad4-independent expression of some Nodal target genes is sub-optimal, but the embryo dynamically compensates for this reduced signaling as a consequence of relieved feedback inhibition from the Nodal antagonists Lft1/2. While this results in sufficient Nodal signaling for axial and paraxial mesoderm induction, endoderm specification is more severely affected. In contrast, early BMP signaling absolutely requires Smad4 to induce target gene expression and promote D/V patterning. Using calibrated doses of a BMP inhibitor and mutant embryos we take advantage of Optical Projection Tomography (OPT), followed by semi-automated segmentation and quantitation, to establish a BMP morphospace. Consistent with our phenotypic findings, this analysis demonstrates that loss of BMP signaling dominates the morphogenetic outcome of MZsmad4a embryos. Taken together, our data indicate that Smad4 is dispensable for Nodal signaling but is essential for BMP signaling and confers robustness to embryo morphogenesis.

## Results

**Smad4a is essential for embryo development.** Due to an ancestral genome duplication, there are two copies of smad4 within the zebrafish genome: smad4a (ENSDARG00000075226) and smad4b (ENSDARG00000012649). We characterized the temporal expression of the two paralogs by qPCR across blastula stages (64-cell to sphere stage), gastrulation (30 to 90% epiboly), somitogenesis and larval stages (24 h post fertilization (hpf)–96 hpf) (Fig. 1a, Supplementary Fig. 1a). Strikingly, the expression of smad4b was very low at all stages examined, whereas smad4a was robustly expressed prior to, during and after gastrulation (Fig. 1a, Supplementary Fig. 1a). Notably, smad4a transcripts were strongly detected between the 64-cell and sphere stages, prior to zygotic genome activation, showing that the transcript is maternally deposited (Fig. 1a). To confirm the maternal origin of the smad4a transcript and investigate its distribution in the embryo we investigated smad4a expression at sphere stage using

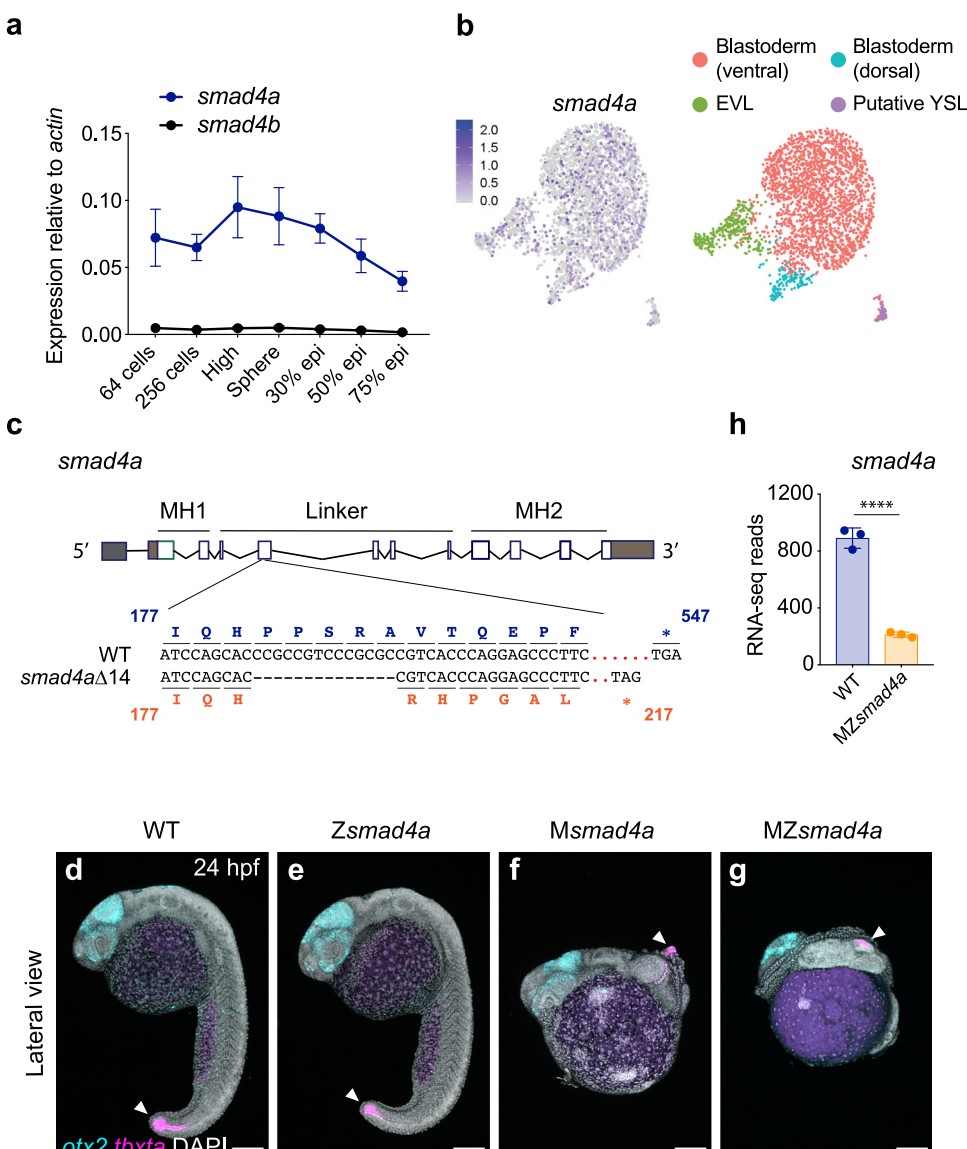

**Fig. 1 smad4a is expressed during early zebrafish development and its loss disrupts embryo patterning. a** qPCR for *smad4a* and *smad4b* mRNA in WT embryos, at the stages indicated. Means ± SEM are shown for four biological replicates for each stage. **b** Uniform manifold approximation and projection (UMAP) visualization of single cells derived from sphere stage zebrafish embryos. Left panel, normalized counts for *smad4a* expression. Right panel, unsupervised clustering subdivides the sphere sample in four different clusters. **c** Schematic representation of the *smad4a* locus and CRISPR/Cas9 editing strategy, with WT and the mutant DNA and protein sequences. **d**–**g** Lateral views of 24-hpf WT (**d**), Z*smad4a* (**e**), M*smad4a* (**f**), and MZ*smad4a* (**g**) embryos double FISH-stained for *otx2* and *tbxta*. Images in **d**–**g** are representative of 25 embryos each and four independent experiments. Scale bar corresponds to 150 μm and arrow indicates the tailbud. Nuclei were stained with DAPI (white). **h** RNA-seq reads for *smad4a* in WT and MZ*smad4a* mutants. Means ± SD are shown for three biological replicates for each genotype. $p(adj) = 2.132 \times 10^{-35}$. Wald test. ****, $p(adj) < 0.0001$.

scRNA-seq (Fig. 1b). Cluster analysis of the sphere library revealed *smad4a* expression within the putative yolk syncytial layer (YSL), the enveloping layer (EVL), and the ventral and dorsal blastoderm (Fig. 1b and see Supplementary Fig. 2a for cluster definitions). This was similar to the expression pattern of *smad5* and *smad2* which are maternally deposited[44,45] (Supplementary Fig. 2b, c). We therefore concluded that *smad4a*, but not *smad4b*, is the major *smad4* during zebrafish embryo development and therefore the gene to target for downstream functional analysis.

To delete *smad4a*, we used a CRISPR/Cas9 genome editing approach[46,47] and generated a *smad4a* null allele, *smad4aΔ14* carrying a 14-bp deletion within the Smad4 linker region which leads to a deleterious frameshift and a premature stop codon (Fig. 1c). Homozygous zygotic *smad4a* mutants (hereafter

Z*smad4a*) generated by in-crossing heterozygous carriers were indistinguishable from WT embryos at 24 hpf (Fig. 1d, e, Supplementary Fig. 1c, d), reached adulthood at the expected mendelian ratio and were fertile (Supplementary Fig. 1b). However, 100% of embryos from homozygous females crossed with either WT, heterozygous or homozygous males died prematurely within 2 days of development (Fig. 1f, g, Supplementary Fig. 1e, f) consistent with a maternal effect. At 24 hpf, mutant embryos displayed a shortened axis and disrupted patterning, with anterior structures abnormally expanded, as shown by expression of anterior (*otx2*) and posterior (*tbxta*) gene landmarks (Fig. 1d–g, Supplementary Fig. 1c–f). These defects appeared more severe in embryos originating from crosses between homozygous female and homozygous males (MZ*smad4a* mutants) (Fig. 1g, Supplementary Fig. 1f), compared to crosses

with either WT or heterozygous males (maternal (M) smad4a mutants) (Fig. 1f, Supplementary Fig. 1e), suggesting a partial rescue by the paternal WT allele. Maternal deposition of smad4a was lost in both MZsmad4a and Msmad4a mutants (from 64 cells to sphere; Supplementary Fig. 1g), likely due to nonsense mediated decay (NMD) within the maternal germline[48]. Similarly, zygotic smad4a levels were also markedly reduced in MZsmad4a mutants confirmed by both qPCR and bulk RNA-seq (Fig. 1h, Supplementary Fig. 1h). It is commonly reported that the deletion of a gene is often functionally compensated by upregulation of its duplicated copy[48]. However, transcript levels for smad4b remained unchanged in both Msmad4a and MZsmad4a mutants as assessed by both qPCR and RNA-seq (Supplementary Fig. 1h, i), which argues against a compensatory effect via smad4b. The mutant phenotype could be fully rescued by injecting either human SMAD4 mRNA or zebrafish smad4a mRNA proving that the phenotype is specific to loss of smad4a (Supplementary Fig. 1j, k). Taken together our results show that the mutant phenotype is dependent on the specific loss of smad4a and demonstrate that smad4a is essential for embryo development.

**Loss of smad4a differentially affects expression of BMP and Nodal target genes.** Having generated a specific smad4a mutant we next investigated how loss of Smad4a influenced Nodal- and BMP-induced transcription. We initially explored BMP and Nodal transcriptional outputs at mid gastrulation, a time at which these ligands pattern the embryo through well-described arrays of target genes[49,50]. We performed RNA-seq on WT and MZsmad4a embryos at 50% epiboly and compared three biological replicates for each condition (Fig. 2a). We found that gene expression was consistently affected across the different embryo clutches (Fig. 2a). By applying a cut-off threshold on the RNA-seq reads (p value-adjusted <0.01) we identified a cohort of 345 genes that were significantly downregulated and 302 genes significantly upregulated upon loss of smad4a (Fig. 2a). Strikingly, we found that most known BMP targets were within the cohort of downregulated genes (Fig. 2b; Supplementary Data File 1), amongst them were sizzled, smad1, eve1, bmp4, and id3[50–52] (Fig. 2b). These findings were corroborated by examining the expression of the BMP targets eve1 and dlx3b[53,54] in a broader temporal window using qPCR. Consistent with the RNA-seq data, expression of these genes was strongly downregulated during gastrulation in MZsmad4a mutants compared to WT embryos (Supplementary Fig. 3a). In sharp contrast to the BMP target genes, we found that at 50% epiboly the expression of Nodal target genes in MZsmad4a embryos was maintained at levels comparable to WT (Fig. 2b; Supplementary Data File 1). This was exemplified by the expression of target genes lft1, gsc, mixl1, ndr2, and dusp4 (Fig. 2b) whose expression strongly depends on Nodal[32–34]. Given this unexpected result we further explored the expression of the Nodal target genes ndr1/2 and lft1/2 from blastula stages to the end of gastrulation. Consistent with the RNA-seq data, at 50% epiboly the expression levels of ndr1 and ndr2 were similar to WT for both Msmad4a and MZsmad4a embryos (Fig. 2c, Supplementary Fig. 3a). Similarly, their expression was unaffected across early (sphere–30% epiboly) and mid (50–75% epiboly) gastrulation stages (Fig. 2c). However, expression of lft1 and lft2 in the MZsmad4a mutants was significantly reduced during early gastrulation stages, although reached WT levels from mid gastrulation onwards (Fig. 2c, Supplementary Fig. 3a). Thus, for Nodal signaling, loss of Smad4a reduces the expression of some target genes, whilst others are completely unaffected.

Given the differential requirement for smad4a in Nodal- and BMP-dependent transcription, we asked how MZsmad4a

mutants would compare to a BMP mutant in terms of gene expression. For this we analyzed a published RNA-seq dataset for bmp7[-/-] mutants at 8 hpf, which lack all zygotic BMP signaling[52,55] and compared the log2 fold change (log2FC) expression for known Nodal and BMP target genes. Global changes in Nodal and BMP target gene expression were equivalent across the two genotypes, further showing that MZsmad4a mutants transcriptionally phenocopy a complete loss of zygotic BMP signaling (Supplementary Fig. 3b).

Having established the requirement for Smad4a in mediating Nodal- and BMP-dependent transcription in zebrafish embryos, we investigated whether these findings could be generalized to other systems. We therefore deleted SMAD4 in mouse embryonic stem cells (mESC) (Supplementary Fig. 4a, b) and analyzed the expression of Nodal and BMP target genes upon Activin A and BMP4 stimulation, respectively (Supplementary Fig. 4c, d). Consistent with our findings in zebrafish, expression of three developmental Nodal target genes Lefty1, Lefty2, and T was either delayed or unaffected in two SMAD4 knockout mESC clones, compared to WT mESCs (Supplementary Fig. 4c), whilst expression of the BMP target genes Id1, Id2, and Id3 was completely abolished in SMAD4-deleted cells (Supplementary Fig. 4d).

From these data we conclude that loss of Smad4 severely inhibits BMP signaling, whilst it has a much less dramatic effect on Nodal signaling.

**Smad4 is required for establishing the initial pSmad1/5 D/V gradient.** Focusing first on the prominent role of Smad4a in BMP signaling, we asked whether Smad4 was essential for recruiting activated pSmad1/5–Smad4 complexes to DNA. We reasoned that this might be the case since the well-characterized binding sites for pSmad1/5–Smad4 complexes (GRCGNC-N$_5$-GTCT), contain binding motifs for both the Smad1/5 Mad homology 1 (MH1) domains and the Smad4 MH1 domain[56–59]. As these binding sites have been well conserved during evolution[12,60,61], we used a mammalian tissue culture system to address this, utilizing a DNA pulldown (DNAP) assay to measure DNA binding to the upstream enhancer region (as defined by a pSMAD1/5 ChIP-seq) of the BMP target gene ID3 in human cells[62]. pSMAD1/5 and SMAD4 readily bound to this enhancer when extracts from BMP4-induced HaCaT cells were used, but not when using extracts from untreated cells (Fig. 3a). By contrast, pSMAD1/5 was unable to bind DNA when extracts from two separate clones of SMAD4 knockout HaCaT cells were used (Fig. 3a). This clearly demonstrates that SMAD4 is required for recruitment of BMP-activated pSMAD1/5–SMAD4 complexes to DNA, which would explain its essential role in mediating BMP responses.

We then went on to investigate when in early embryogenesis the requirement for Smad4a for BMP signaling was first evident, determining whether the BMP signaling gradient, which is established by 40% epiboly[41,63], was affected in the MZsmad4a mutants (Fig. 3b; Supplementary Fig. 5a). To exclude possible differences in signaling levels dependent on the embryo background, we used crosses between heterozygous females with homozygous males as a control (hereafter CTRL), which are indistinguishable from WT (Fig. 1d, e, Supplementary Fig. 1c, d). At 40% epiboly, CTRL and WT embryos displayed graded pSmad1/5 activity across the D/V axis, whereas in MZsmad4a mutants the pSmad1/5 gradient was absent, apart from a few pSmad1/5-positive cells at the very ventral margin of the embryo (Fig. 3b, c; Supplementary Fig. 5a, b). This indicated that in the absence of Smad4a, almost all BMP signaling activity was abolished. The lack of pSmad1/5 in the MZsmad4a embryos

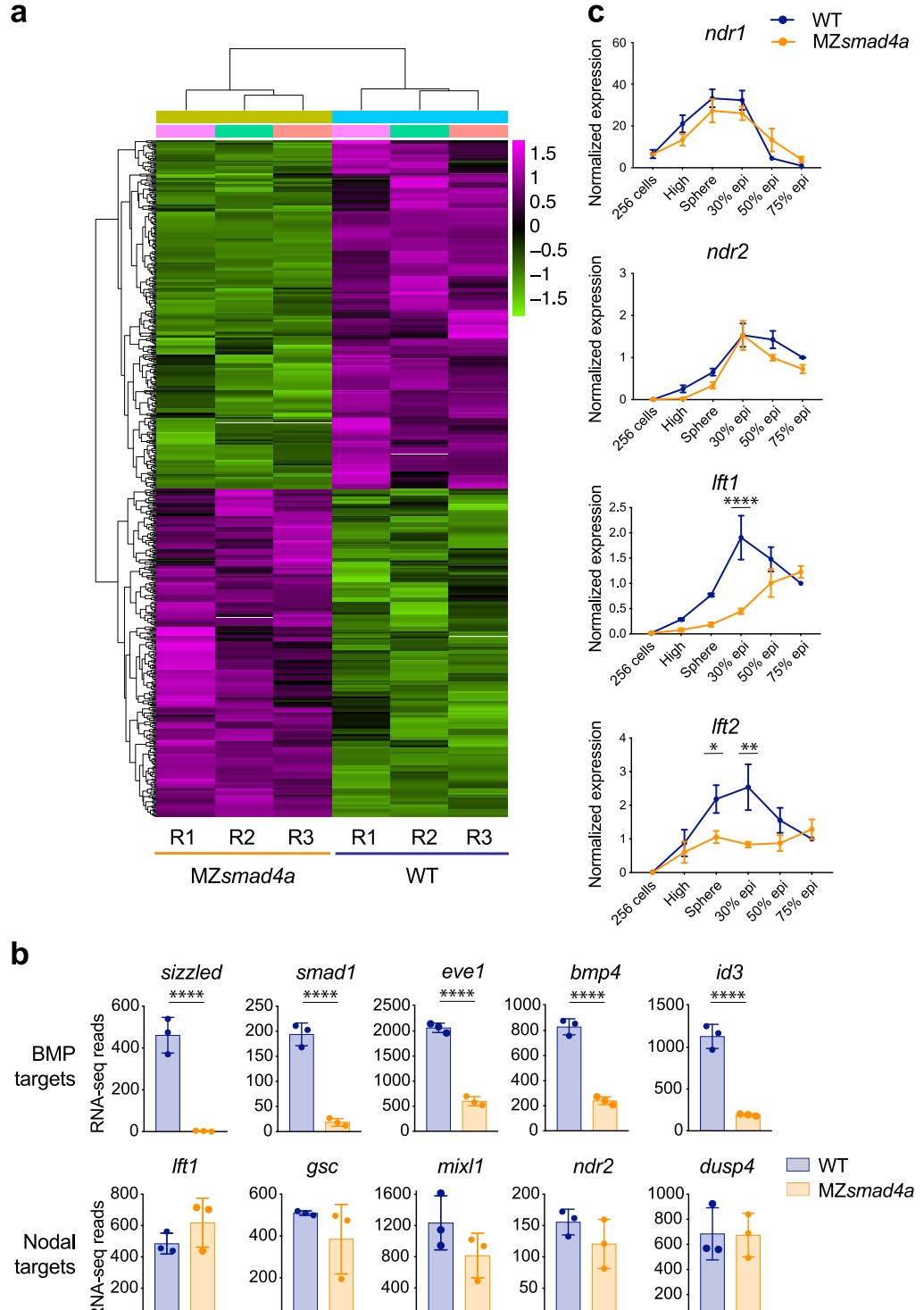

**Fig. 2 BMP-dependent transcription is abolished in *MZsmad4a* mutants, while Nodal target gene expression is maintained. a** RNA-seq on WT and *MZsmad4a* embryos performed in triplicate. In the heatmap the color legend shows scaled RLog transformed expression counts for each replicate. **b** RNA-seq reads for representative BMP (upper panel) and Nodal (bottom panel) target genes in WT and *MZsmad4a* embryos. The data are means ± SD from three biological replicates for each genotype. *sizzled*: $p(\text{adj}) = 4.699 \times 10^{-78}$, *smad1*: $p(\text{adj}) = 9.253 \times 10^{-23}$, *eve1*: $p(\text{adj}) = 1.581 \times 10^{-27}$, *bmp4*: $p(\text{adj}) = 2.183 \times 10^{-23}$, *id3*: $p(\text{adj}) = 2.445 \times 10^{-48}$. Wald test. ****$p(\text{adj}) < 0.0001$. **c** qPCR for *ndr1/2 and lft1/2* in WT and *MZsmad4a* embryos at the indicated stages. Normalized values are shown as means ± SEM. For *ndr1/2*, the data are the result of three biological replicates while for *lft1/2* they are the result of four biological replicates, except for *lft1* in *MZsmad4a* embryos at sphere stage and *lft2* in WT embryos at 30% epiboly, which are the result of three biological replicates. *lft1* 30% epiboly: $p(\text{adj}) = 4389 \times 10^{-5}$, *lft2* sphere: $p(\text{adj}) = 0.046$, *lft2* 30%: $p(\text{adj}) = 0.002$. Unpaired multiple comparison $t$-test with Holm-Sidack correction. *$p < 0.05$; **$p < 0.01$; ****$p < 0.0001$.

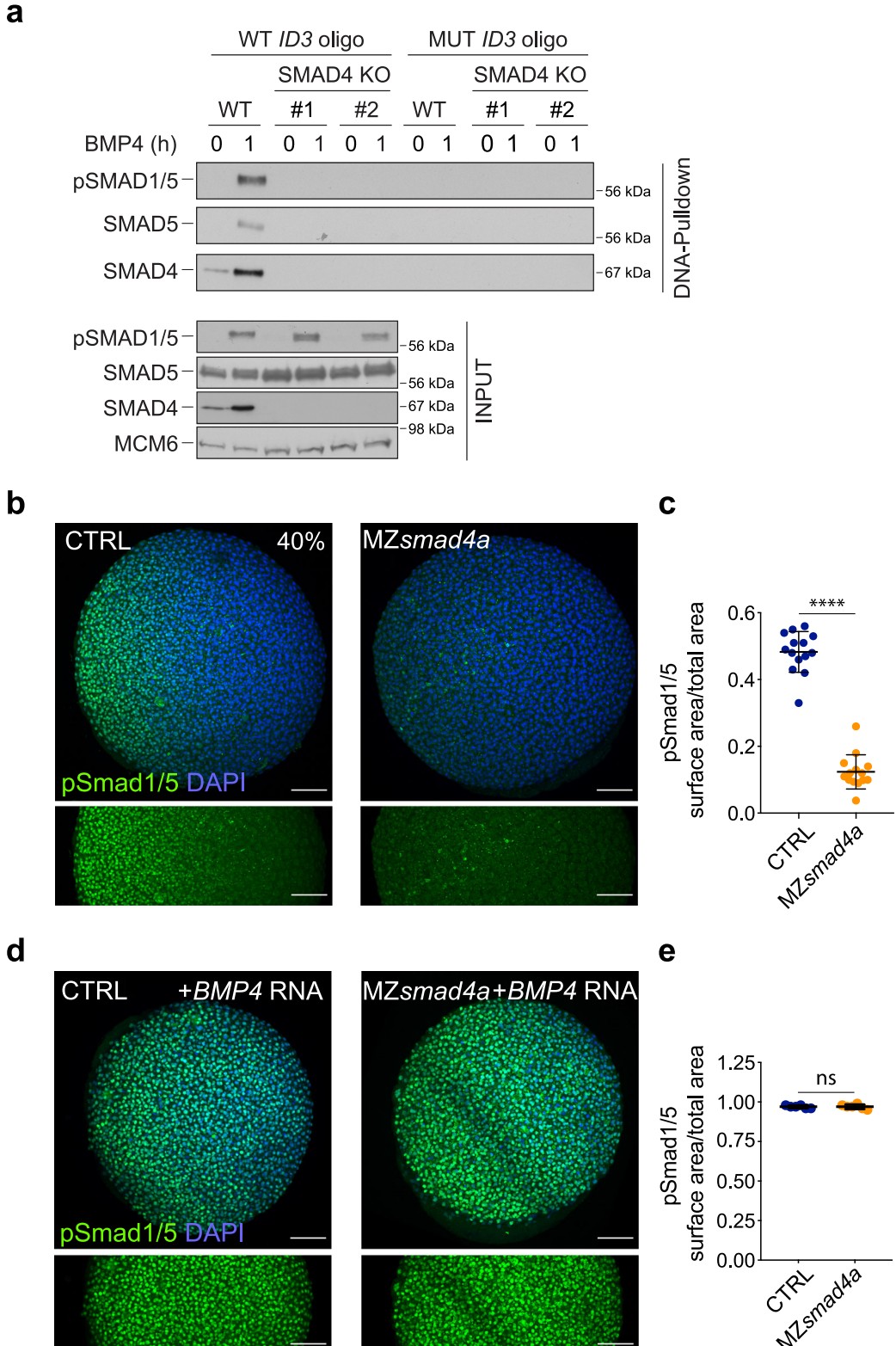

was initially surprising, given that Smad4 acts downstream of receptor-induced Smad1/5 phosphorylation[64]. However, we reasoned that this might reflect an inability of MZ*smad4a* embryos to produce zygotic BMP ligands, a process that is known to require BMP signaling itself via the ligand Gdf6a (also called radar)[65]. Alternatively, it could reflect a loss of BMP receptors or other pathway components in the MZ*smad4a* mutants. To test these hypotheses, we determined whether injected *BMP4* mRNA

could rescue pSmad1/5 activity in MZ*smad4a* embryos. Indeed, ectopic ligand expression induced widespread pSmad1/5 activity in both CTRL and MZ*smad4a* embryos (Fig. 3d, e), proving that the BMP signaling machinery is present, but cannot be activated in MZ*smad4a* mutants due to defective ligand production downstream of Smad4a. Further confirming this result, maternal deposition of *smad5* and *alk8* is unaffected in MZ*smad4a* embryos (Supplementary Fig. 2c). Importantly, BMP4 overexpression was

**Fig. 3 BMP/pSmad1/5 activity is lost in MZ*smad4a* mutants but can be rescued by ectopic BMP ligand expression. a** WT HaCaT cells and two clones of SMAD4 knockout (KO) HaCaT cells were treated or not with BMP4. DNA pulldowns from whole cell extracts were performed using an oligonucleotide containing the SMAD1/5–SMAD4 binding sites of the upstream *ID3* enhancer, or a version in which these sites were mutated. Pulldowns were immunoblotted with the indicated antibodies and inputs are shown below. The immunoblot is representative of three independent experiments. Molecular weight markers are given in kDa on the right of the blots. **b** Immunostaining for pSmad1/5 (green) in 40% epiboly CTRL and MZ*smad4a* embryos. Nuclei are stained with DAPI (blue). The bottom panel highlights the pSmad1/5 gradient without the DAPI channel. Scale bars correspond to 100 μm. Animal views are shown. **c** Quantitation of the pSmad1/5 gradient size with respect to the embryo surface. 14 embryos for each group are represented as means ± SD. $p = 4.985 \times 10^{-8}$. Two sided Mann–Whitney test. ****$p < 0.0001$. **d** As in **b** but showing CTRL and MZ*smad4a* embryos injected with 60 pg of h*BMP4* mRNA. Scale bars correspond to 100 μm. **e** Quantitation of h*BMP4* mRNA-injected CTRL and MZ*smad4a* embryos. Six embryos for each group are represented as means ± SD. Two sided Mann–Whitney test. ns not significant.

unable to phenotypically rescue the MZ*smad4a* embryos (Supplementary Fig. 5c, d), indicating that Smad4a is required for BMP-induced transcriptional responses and not just for zygotic BMP ligand production (see Discussion).

**Nodal signaling is functional in the absence of Smad4a.** The finding that at 50% epiboly expression of Nodal target genes was similar in WT and MZ*smad4a* mutant embryos was unexpected. We therefore wanted to confirm that the expression of Nodal target genes in the MZ*smad4a* mutants was due to Nodal signaling and not to another compensating signaling pathway. Thus, we ascertained whether expression of these target genes was sensitive to inhibition by the Nodal type I receptor inhibitor, SB-505124[66]. Indeed, expression of *lft1/2* and *ndr2* in MZ*smad4a* mutants was fully dependent on Nodal signaling, as pathway inhibition with SB-505124 completely abolished gene expression as assessed by qPCR and in situ hybridization (ISH) (Fig. 4a–c; see Fig. 5b), thus excluding any input from other signaling pathways. We next wanted to eliminate the possibility that very low levels of Smad4b could account for expression of Nodal target genes in MZ*smad4a* mutants. To address this, we used a *smad4b* morpholino (MO). Injection of the *smad4b* MO into MZ*smad4a* embryos had no effect on the expression of *lft1* (Supplementary Fig. 3c). We could demonstrate that the MO was specific, because while injection of *smad4b* mRNA with a *smad4b* miss-paired (mp) MO rescued the MZ*smad4a* mutant phenotype, injection of *smad4b* mRNA in combination with the *smad4b* MO could not (Supplementary Fig. 3d). Thus, we conclude that Smad4b does not compensate for the lack of Smad4a in mediating Nodal responses.

To further characterize Nodal signaling activity in MZ*smad4a* embryos, we investigated in detail the spatial expression pattern of *lft1* and *ndr1* at the mid-gastrula stage. We and others have previously shown that *ndr1* and *ndr2* are initially expressed in the YSL, and then promote their own expression, as well as expression of *lft1/2*, in adjacent blastomeres to progressively extend up to five cell tiers by mid gastrulation[32,33,67]. By recording nuclear distances from the embryo margin, we found that at 50% epiboly, *lft1* was expressed in a domain of the same dimensions and at equivalent levels in CTRL and MZ*smad4a* mutant embryos (Fig. 4a–c). We observed that *ndr1* was expressed by the YSL and blastomeres at the margin of both MZ*smad4a* mutant and CTRL embryos, and the domain size of about five cell tiers was equivalent in both cases (Fig. 4d–g). We noted that the expression of *ndr1* in the nuclei of MZ*smad4a* mutants appeared consistently higher than in the CTRLs (Fig. 4d–g). As this represents newly synthesized transcripts, it suggested that ongoing synthesis of *ndr1* is higher at this stage in the mutants compared with the controls.

**Smad4-independent Nodal signaling is less robust.** We were struck by the fact that at mid gastrulation (50% epiboly), Nodal-induced gene expression in MZ*smad4a* embryos was nearly

equivalent to that in WT embryos and that the Nodal signaling gradient extended up to five cell tiers in both WT and mutant embryos. However, at earlier time points *lft1/2* were clearly delayed in the MZ*smad4a* mutants compared with their WT counterparts (see Fig. 2c; Supplementary Fig. 3a). We reasoned that as Lft1/2 are known to regulate Nodal signaling activity via a negative feedback mechanism, it was possible that Nodal-induced transcription was suboptimal in MZ*smad4a* embryos but managed to compensate by mid gastrulation because of the delay in Lft1/2-mediated inhibition. To explore this possibility, we investigated the pSmad2/3 gradient at the embryonic margin as a more direct readout of Nodal signaling activity. Signal transduction of the Nodal arm of the TGF-β pathway is typically mediated by activated SMAD2 and SMAD3[68]. However, in zebrafish, only Smad2 is expressed at early blastula and gastrula stages (Supplementary Fig. 2b)[33]. We could readily detect nuclear accumulation of pSmad2 in both CTRL and MZ*smad4a* embryos, consistent with data in SMAD4-null cell lines showing that SMAD4 is not required for nuclear accumulation of phosphorylated R-SMADs[69]. Strikingly however, compared to CTRL embryos, pSmad2 levels were more intense and pSmad2 prematurely extended up to five cell tiers in 40% epiboly MZ*smad4a* mutant embryos (Fig. 4h, i, Supplementary Fig. 5e). This was also observed when MZ*smad4a* mutants were compared to WT embryos (Supplementary Fig. 5f).

Given that in other developmental systems there is well-documented antagonism between Nodal and BMP signaling (for an example, see ref. [70]), we asked whether this enhanced Nodal signaling was a consequence of the loss of BMP signaling in the mutant embryos. However, this was not the case, as BMP signaling inhibition via the receptor inhibitor DMH1[71] did not affect the pSmad2 gradient at 40% epiboly (Supplementary Fig. 5g, h). It is therefore likely that the pSmad2 gradient spreads prematurely in the mutants as a result of reduced expression of Lft1 and Lft2. This would fit well with the behavior of the pSmad2 gradient in *lft1*[-/-];*lft2*[-/-] double mutants, compared with WT embryos[35]. We therefore propose that in the absence of Smad4, pSmad2 homotrimers are recruited by transcription factors such as Foxh1 and Mixl1 bound to Nodal-regulated enhancers and can activate transcription of target genes like *lft1/2*, albeit less efficiently than in the WT context (Fig. 5a). This is consistent with the findings that the transcription factors Foxh1 and Mixl1 are essential for Nodal responses in zebrafish[72,73], and also with our previous data showing that SMAD4 is not necessary for transcriptional activation if the pSMAD2 homomeric complex is recruited to chromatin via another transcription factor[74]. Thus, during blastula and early gastrula stages when Lft1/2 levels are low in the MZ*smad4a* mutants, the pSmad2 gradient grows faster than in the WT embryos. This in turn promotes expression of target genes, including those encoding Lft1/2, which then act to prevent the pSmad2 gradient growing further (Fig. 5a).

It has been noted by others that Lft1 and Lft2 provide robustness to Nodal signaling in zebrafish embryos[35]. Given that *lft1/2* expression is initially compromised in early MZ*smad4a*

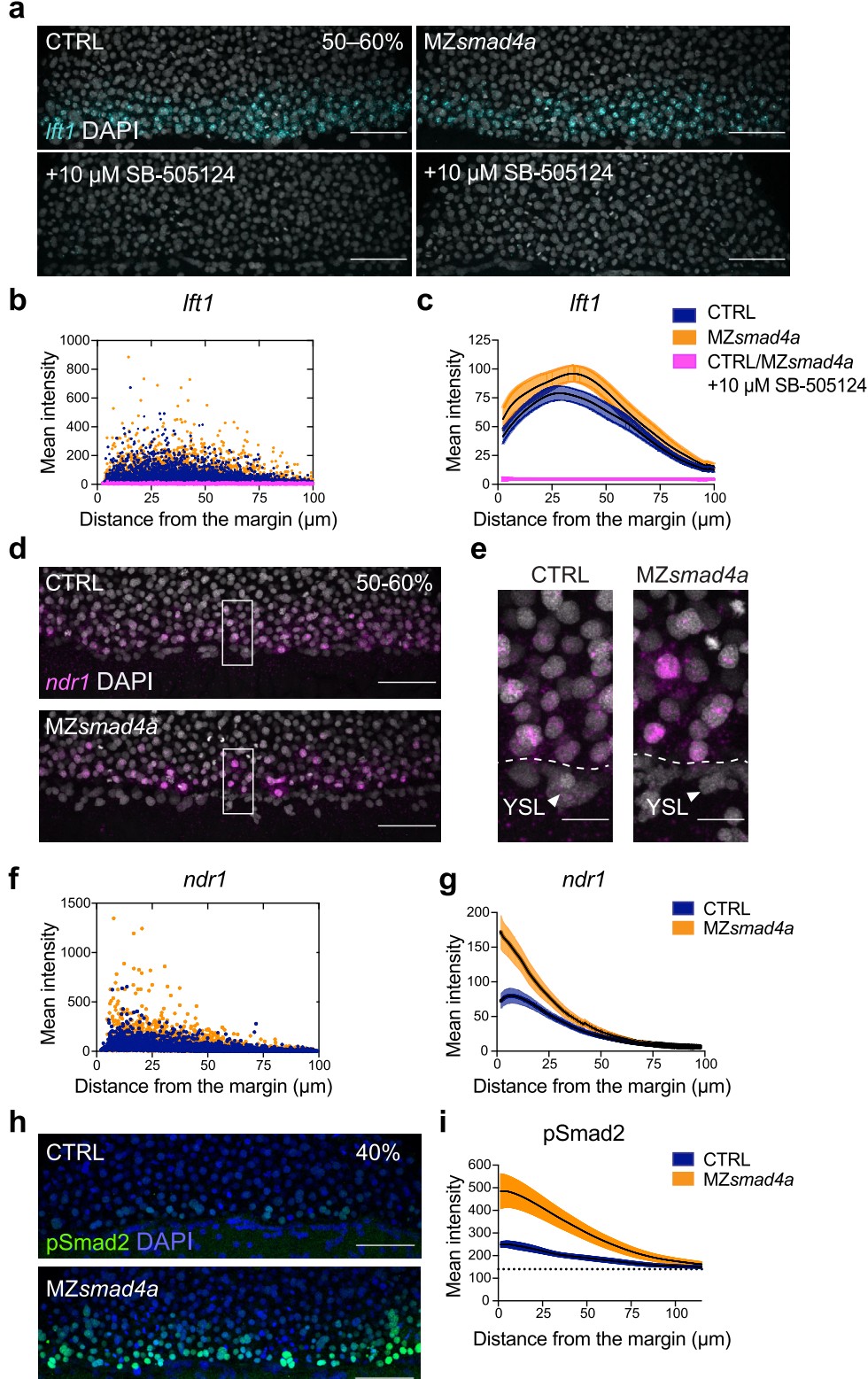

mutants, we surmised that Smad4-independent Nodal signaling may also be less robust than that in WT embryos. A test of robustness is the sensitivity of a system to perturbation. Therefore, we performed titration experiments using increasing doses of Nodal receptor inhibitor SB-505124 and quantified gene expression responses downstream of Nodal by qPCR at 50% epiboly. Compared to WT embryos, expression of *ndr2* or *lft1* in MZ*smad4a* embryos appeared markedly sensitized, being

significantly lower than WT at low doses of inhibitor. The effect on *lft2* was less strong but still followed a similar trend (Fig. 5b). Thus, in the absence of Smad4a, gene expression downstream of Nodal–pSmad2 signaling is more fragile.

**Smad4-independent Nodal signaling is sufficient for mesoderm induction.** Having shown that, despite lacking robustness, a Nodal signaling gradient forms at the margin of MZ*smad4a*

**Fig. 4 The Nodal–pSmad2 signaling gradient forms independently of Smad4a. a** Maximum intensity projection (MIP) of quantitative ISH for *lft1* at mid gastrulation (50–60% epiboly) in CTRL and MZ*smad4a* embryos treated ±10 μM SB-505124 at sphere stage. Scale bars correspond to 80 μm. Lateral views are shown. Nuclei were stained with DAPI (white). **b** Quantitation of **a**. Orange, dark blue, and magenta dots show segmented cells from CTRL and MZ*smad4a* embryos treated ±10 μM SB-505124. *n* = 8 embryos for each group. **c** Curve fitting for segmented cell intensities shown in **b**. Means are shown by black lines and the orange/dark blue shading indicates the SEMs. **d**, **e** MIP of quantitative ISH for *ndr1* in CTRL and MZ*smad4a* embryos at mid-gastrulation (50–60% epiboly). Regions indicated in **d** by white boxes are shown enlarged in **e**. Scale bars correspond to 80 μm in **d** and 20 μm in **e**. Arrowheads indicate YSL. Lateral views are shown. Nuclei were stained with DAPI (white). **f** Quantitation of **d**. Orange and dark blue dots show segmented cells from CTRL and MZ*smad4a* embryos. *n* = 8 embryos for each group. **g** Curve fitting for segmented cell intensities shown in **f**. Means are shown by black lines and the orange/dark blue shading indicates the SEMs. **h** MIP of CTRL and MZ*smad4a* at 40% epiboly, immunostained for pSmad2. Scale bar corresponds to 80 μm. Lateral views are shown. Nuclei were stained with DAPI (blue). **i** Quantitation of the pSmad2 immunostaining at the margin of CTRL and MZ*smad4a* embryos. Each trace represents *n* = 8 embryos. Means are shown by black lines and the orange/dark blue shading indicates the SEMs. Dotted line shows baseline.

embryos, which is sufficient to induce the expression of Nodal target genes, we asked whether Smad4-independent Nodal signaling was sufficient for inducing mesoderm and endoderm in MZ*smad4a* embryos. We therefore investigated the expression of endoderm and axial/paraxial mesoderm progenitors, between mid and late gastrulation, the time at which they are being progressively specified within separated domains in the embryo. As shown by ISH, expression of the axial mesoderm marker *tbxta* was similarly expressed on the dorsal side of both WT and MZ*smad4a* mutant embryos (Fig. 5c). However, it was fainter at the edges of the transcriptional domain in the mutants. Despite this, overall transcript levels were equivalent as shown by RNA-seq and qPCR (Supplementary Fig. 6a, b). Similarly, expression levels of the axial mesoderm marker *chrd* and the notochord marker *noto* were also unaffected (Supplementary Fig. 6b, c), as was expression of the paraxial mesoderm markers *tbx16*, *mespaa*, and *mespab* (Fig. 5c; Supplementary Fig. 6a). This shows that both axial and paraxial mesoderm are normally induced in MZ*smad4a* embryos. In contrast to this finding, we observed a reduced number of endodermal progenitors, marked by *sox32* (Fig. 5d). This was also corroborated by a significant reduction in *sox17* (Supplementary Fig. 6a), which is downstream of Sox32 within the endodermal lineage[75]. For a complete picture of germ layer specification, we also investigated ectodermal derivatives which are regulated by BMP signaling. Interestingly, expression of anterior neural plate markers like *otx1*, *six3a*, and *sox2*, which are typically antagonized by BMP signaling[76] was unchanged or slightly upregulated (Supplementary Fig. 6a), while expression of ventral epidermal markers like *gata2* and *foxi1* which are induced by BMP[77,78] was downregulated (Supplementary Fig. 6a). Interestingly, these changes in D/V gene expression were recapitulated in *bmp7*[−/−] mutants. However, in contrast to MZ*smad4a* mutants, expression of endodermal markers was largely unaffected in *bmp7*[−/−] mutants (Supplementary Fig. 6a). Therefore, while BMP-induced ectodermal derivatives are severely compromised in MZ*smad4a* mutants, Smad4-independent Nodal signaling is sufficient for mesoderm induction, although not for optimal endoderm specification.

**Loss of Smad4a steers embryo development towards a BMP zero morphotype.** Loss of BMP signaling dominates the transcriptional and signaling landscape in MZ*smad4a* embryos, while transcription downstream of Nodal is relatively normal. As embryo morphology is the result of these early signaling interactions, we asked if MZ*smad4a* mutants would also be morphologically comparable to BMP mutants. We first looked at global developmental dynamics using light sheet microscopy to image WT and MZ*smad4a* embryos from mid-gastrulation until 20 hpf (Supplementary Movies 1 and 2). While epiboly proceeded normally in MZ*smad4a* embryos, by the end of gastrulation MZ*smad4a* mutants were abnormally elongated (Fig. 6a).

Moreover, convergence extension (C/E) at the dorsal side occurred in a disorganized way in MZ*smad4a* embryos, resulting in a poorly patterned axis (Fig. 6a; Supplementary Movie 2). Furthermore, a proportion of cells failed to undergo dorsal migration. Instead, they formed clusters on the ventral side of the embryo (Fig. 6a; Supplementary Movie 2).

Both embryo elongation and C/E defects have been reported in *bmp* mutants[79]. To investigate the similarity between the morphology of MZ*smad4a* mutants and *bmp* mutants more rigorously, we developed a method to image large numbers of embryos and to objectively and quantitatively describe their morphological features. To achieve this we took advantage of an OPT system, where samples are rotated through 360° with images acquired at set intervals, and a back-projection technique is applied to reconstruct the 3D images[80–82]. The system enabled simultaneous acquisition of up to five embryos (Fig. 6b, Supplementary Movies 3 and 4), This was implemented with a semi-automated pipeline for segmentation and quantitation of 24-hpf fluorescently labeled embryos. Nuclear labeling was used to generate a 3D mask of the embryo, as well as a 3D skeleton endowed within the mask[83]. Anterior and posterior coordinates were defined by the use of *otx2* and *myod* expression landmarks (Fig. 6c–f). These geometrical features were then analyzed to generate an array of classical and ad hoc morphological descriptors (Supplementary Data File 2). Amongst these descriptors were parameters describing differences in embryo thickness (variation of skeleton distance to surface voxels), non-homogeneities of the axis (solidity), and A/P axis shortening (principal axis length ratios, tortuosity, AP index) (Fig. 6g, Supplementary Fig. 7a).

Having developed a sensitive method to describe embryo shape, we next generated a framework describing different extents of BMP perturbation to use as a reference for characterizing MZ*smad4a* mutant morphology. To achieve this, we performed a dose response of increasing concentrations of the BMP receptor inhibitor DMH1 and imaged 4–5 embryos for each dose at 24 hpf (Fig. 7a; Supplementary Fig. 7b). The resulting dataset was then combined with an equivalent dataset generated from a pool of MZ*smad4a*, M*smad4a*, and *bmp2b*[−/−] embryos (the *bmp2b*[tdc24] allele which lacks all zygotic BMP signaling[84]) and visualized using principal component analysis (PCA) (Fig. 7b–e). We found that variation in embryo shape was mainly explained by two principal components (Supplementary Fig. 8a): PC1 (mainly driven by tortuosity and principal axis length ratio L3/L1), and PC2 (driven by solidity) which defined the main axis of the BMP morphospace (Supplementary Fig. 8b, c).

Embryos treated with different doses of DMH1 organized in the morphospace and formed four different clusters according to different degrees of severity, with cluster four being defined by the highest DMH1 concentrations (Fig. 7b, c). Strikingly, M*smad4a* mutants, which display milder morphological defects, fell within

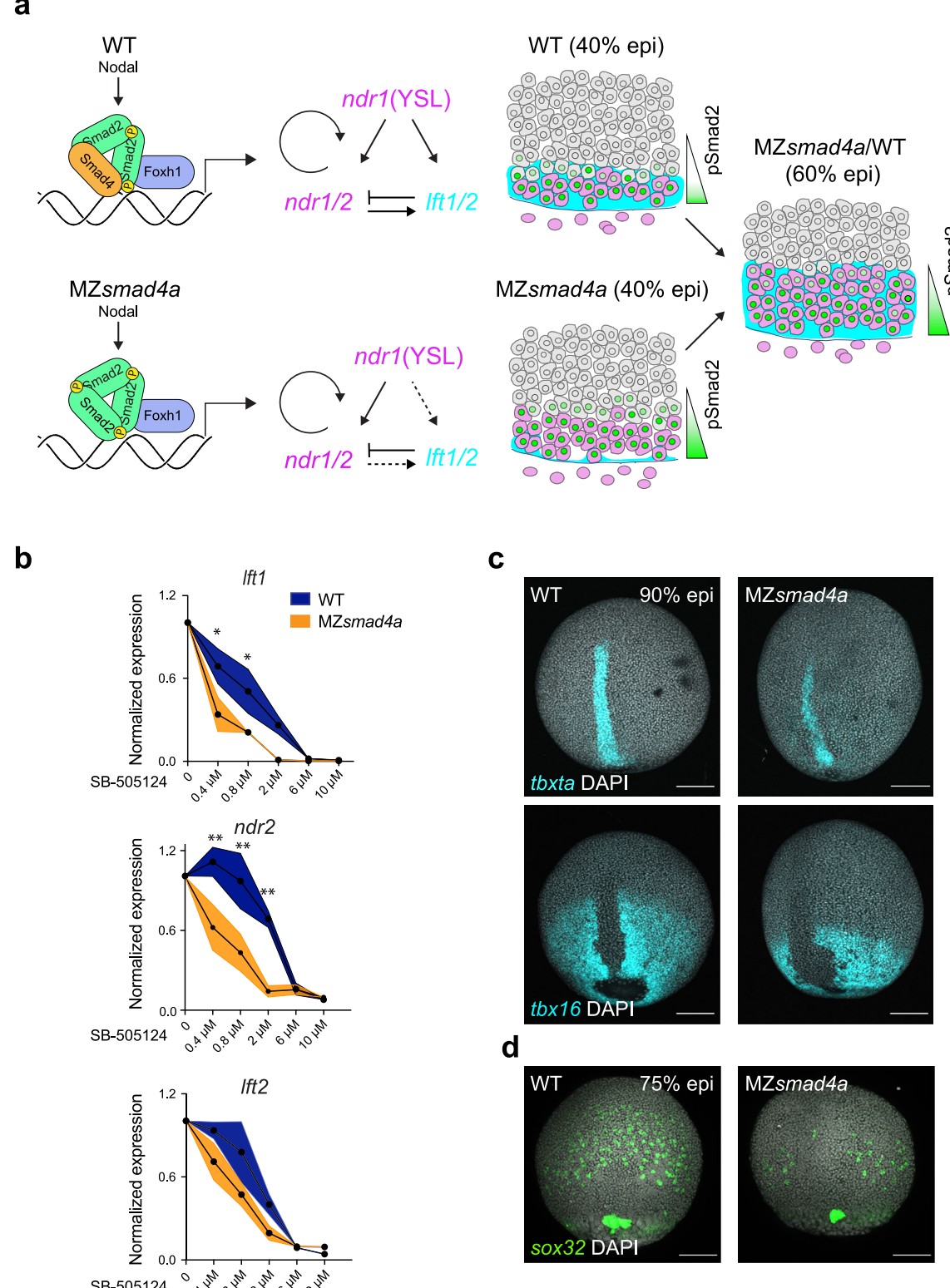

clusters two, three and four (Fig. 7d, e). However, MZ*smad4a* embryos, together with *bmp2b*[-/-] embryos, clustered mainly at the end of the trajectory, within cluster four (Fig. 7d, e). These results indicate that MZ*smad4a* mutants are indistinguishable from embryos in which BMP signaling has been pharmacologically or genetically perturbed.

As a final control, to confirm that our morphometric analysis was specific to mutants compromised for BMP signaling, we performed the same analysis on a distinct signaling mutant, MZ*oep*, which is an MZ mutant for the Nodal co-receptor Tdgf1. This mutant is null for Nodal signaling, and has a phenotype indistinguishable from the *ndr1*[-/-];*ndr2*[-/-] (formerly *sqt;cyc*) double mutant[85]. These embryos clearly did not cluster with either MZ*smad4a*, or *bmp2b*[-/-] embryos (Supplementary Fig. 8d), indicating that the methodology we are employing is capable of distinguishing morphologies arising from inhibition of different signaling pathways.

**Fig. 5 Nodal-independent signaling is sensitized but is sufficient for mesoderm induction. a** Model of Nodal signaling gradient formation in WT and MZ*smad4a* mutant embryos. In contrast to WT (upper panel), in MZ*smad4a* embryos (lower panel) a homomeric trimer of activated pSmad2 can still be recruited to chromatin via Foxh1, but with reduced efficiency. Consequently, some Nodal target genes, in particular, *lft1/2* are initially inefficiently transcribed. As Nodal induction of Ndr1/2 is not inhibited by Lft1/2, this leads to higher premature induction of the Nodal–pSmad2 signaling gradient. By 60% epiboly, once Lft1/2 levels have reached those equivalent to WT, the signaling gradient in MZ*smad4a* mutant embryos has adjusted to WT-like levels. **b** Dose responses of SB-505124 for *lft1/2* and *ndr2* in WT and MZ*smad4a* embryos, treated with inhibitor at sphere stage. The dose marked 0 corresponds to DMSO. Black dots represent the mean for three biological replicates for each dose and each genotype. Orange shading indicates the SEM. *lft1* 0.4 µM: $p(\mathrm{adj}) = 0.012$, *lft1* 0.8 µM: $p(\mathrm{adj}) = 0.036$, *ndr2* 0.4 µM: $p(\mathrm{adj}) = 0.005$, *ndr2* 0.8 µM: $p(\mathrm{adj}) = 0.003$, *ndr2* 2 µM: $p(\mathrm{adj}) = 0.003$. Unpaired multiple comparison *t*-test with Holm-Sidack correction. $*p < 0.05$; $**p < 0.01$. **c** Upper panel: MIP of ISH for *tbxta* (cyan) in CTRL and MZ*smad4a* embryos at 90% epiboly (epi). Bottom panel: MIP of ISH for *tbx16* (cyan) at 90% epiboly. Nuclei are stained with DAPI (white). Scale bars correspond to 150 µm. Dorsal views are shown. **d** MIP of ISH for *sox32* (green) in CTRL and MZ*smad4a* embryos at 75% epiboly. Nuclei are stained with DAPI (white). Scale bars correspond to 150 µm. Dorsal views are shown. Images in **c**, **d** are representative of 25 embryos each and three independent experiments, except for the *tbx16* ISH, which is the result of two independent experiments.

## Discussion

Here we have comprehensively characterized a Smad4 mutant in zebrafish and shown that Smad4 is differentially required for the two arms of the TGF-β family signaling pathways. We have demonstrated this by interrogating Smad4 function downstream of BMP and Nodal, which signal through pSmad1/5 and pSmad2, respectively. By complementing our analysis with an imaging pipeline enabling quantitation of whole 24-hpf embryo morphology, we demonstrate that while Smad4a is essential for BMP–pSmad1/5 signaling, it is dispensable for Nodal–pSmad2 signaling. We also show that loss of SMAD4 in mESCs recapitulates the transcriptional defects observed in zebrafish embryos, indicating that the SMAD4 requirement in mediating BMP and Nodal responses is evolutionarily conserved. We thus conclude that SMAD4 is not an essential component of all TGF-β family signaling pathways. This work has major implications for diseases, in particular, pancreatic and colon cancer, where SMAD4 is frequently mutated or deleted. It will be important in the future to understand in the cancer context, to what extent TGF-β family signaling is still functional in tumors harboring SMAD4 mutations and deletions. Furthermore, because we can rescue our MZ*smad4a* mutants with human *SMAD4* mRNA, we have an ideal system with which to analyze the functional consequences of *SMAD4* mutations found in human tumors.

**Smad4a is essential for BMP signaling**. We show that loss of maternal Smad4 impairs gastrulation and overall embryo development and demonstrate that the severe gastrulation defects exhibited by M*smad4a* and MZ*smad4a* mutants are a consequence of suppressed BMP signaling activity. We provide biochemical, transcriptional, and morphological evidence that Smad4a is essential for pSmad1/5 signaling. Some of the first targets of the BMP signaling pathway in zebrafish embryos are actually the genes encoding zygotic Bmp2b and Bmp7a themselves[65], which we conclude are not produced in the MZ*smad4a* mutant embryos. Consistent with this, we could rescue pSmad1/5 activity at 40% epiboly in the MZ*smad4a* mutants by overexpressing BMP4. Crucially, this was not sufficient to rescue the MZ*smad4a* mutant phenotype, indicating that Smad4a is also required for signaling downstream of Bmp2b and Bmp7a. We noted from our RNA-seq data that at the level of mRNA, *bmp2b*, and *bmp7a* levels were not substantially different between WT and MZ*smad4a* mutants. It is known that the earliest expression of *bmp2b* and *bmp7a* depends on the maternal POU domain transcription factor Pou2/Pou5f1[86], and is not under the control of the BMP pathway[38,41]. *bmp7a* has also been shown to be maternally produced[45]. It would appear that these early *bmp2b/7a* transcripts are not efficiently translated, consistent with the absence of a phenotype in maternal *bmp7a* mutants[37,55]. Further supporting the lack of BMP ligand activity in MZ*smad4a*, we show that *bmp7*⁻/⁻ mutants, which lack all

pSmad1/5 activity[52], phenocopy MZ*smad4a* mutants with respect to the expression of BMP target genes and D/V markers.

Intriguingly, we also show that Z*smad4a* mutants are indistinguishable from WT embryos during early zebrafish development. Moreover, they reach adulthood and are fertile. This suggests that maternally deposited *smad4a* transcripts are sufficient to drive the main signaling and morphogenetic events during Z*smad4a* mutant gastrulation. Consistent with our findings, zygotic *bmp2b* (*swirl*) and *bmp7a* (*snailhouse*) mutants can also reach adulthood if transcripts for these genes are injected at the 1-cell stage[37,39]. However, the contribution of additional BMP ligands, like Bmp4, is not known is this context. Indeed, it is likely that the BMP signaling requirement may be substantial also during juvenile and adult zebrafish life. We have shown that *smad4b* is not expressed up to 4 dpf, nor is it upregulated in the absence of *smad4a*. However, it is possible that at later times Smad4b can mediate BMP signaling instead of Smad4a. Alternatively another transcriptional cofactor may interact with pSmad1/5 to induce transcription of target genes in juveniles and adults instead of Smad4.

**Smad4-independent Nodal signaling is sufficient for mesoderm, but not endoderm induction**. The R-SMADs SMAD1/5 and SMAD3 contact DNA directly via their N-terminal MH1 domains[12]. However, the MH1 domain of SMAD2 does not efficiently bind DNA and thus pSMAD2–SMAD4 complexes are primarily recruited to DNA by other transcription factors, most notably in early embryonic development, Foxh1 and Mixl1 (formerly called Mixer)[12]. Indeed, zebrafish embryos lacking both Foxh1 and Mixl1 resemble embryos devoid of Nodal signaling, albeit less severe[72,73]. We show here that pSMAD1/5 requires SMAD4 for DNA binding, and have previously shown a similar requirement of SMAD4 for pSMAD3 to bind DNA[69]. In endogenous contexts, SMAD4 is found to be a component of activated Foxh1–SMAD complexes and Mixl1–SMAD complexes[14,87]. However, it does not appear to be absolutely required for pSMAD2 binding to these transcription factors, as luciferase reporters driven by Foxh1 or Mixl1 binding sites are ligand-inducible, even in SMAD4-null cells[88,89]. Furthermore, pSMAD2 can induce transcription without the requirement for SMAD4, if it is brought to DNA via a SMAD-interacting transcription factor[74]. This indicates that in the absence of SMAD4, homomeric pSMAD2 complexes can be recruited to enhancers of Nodal target genes by interacting directly with transcription factors like Foxh1 and Mixl1.

Here we demonstrate the functional ramifications of this mechanism in Smad4-null zebrafish embryos. We show that Nodal signaling is induced in MZ*smad4a* embryos independently of Smad4a but appears to be less efficient, affecting the induction of some Nodal targets more than others. We show that levels of *ndr1/2*

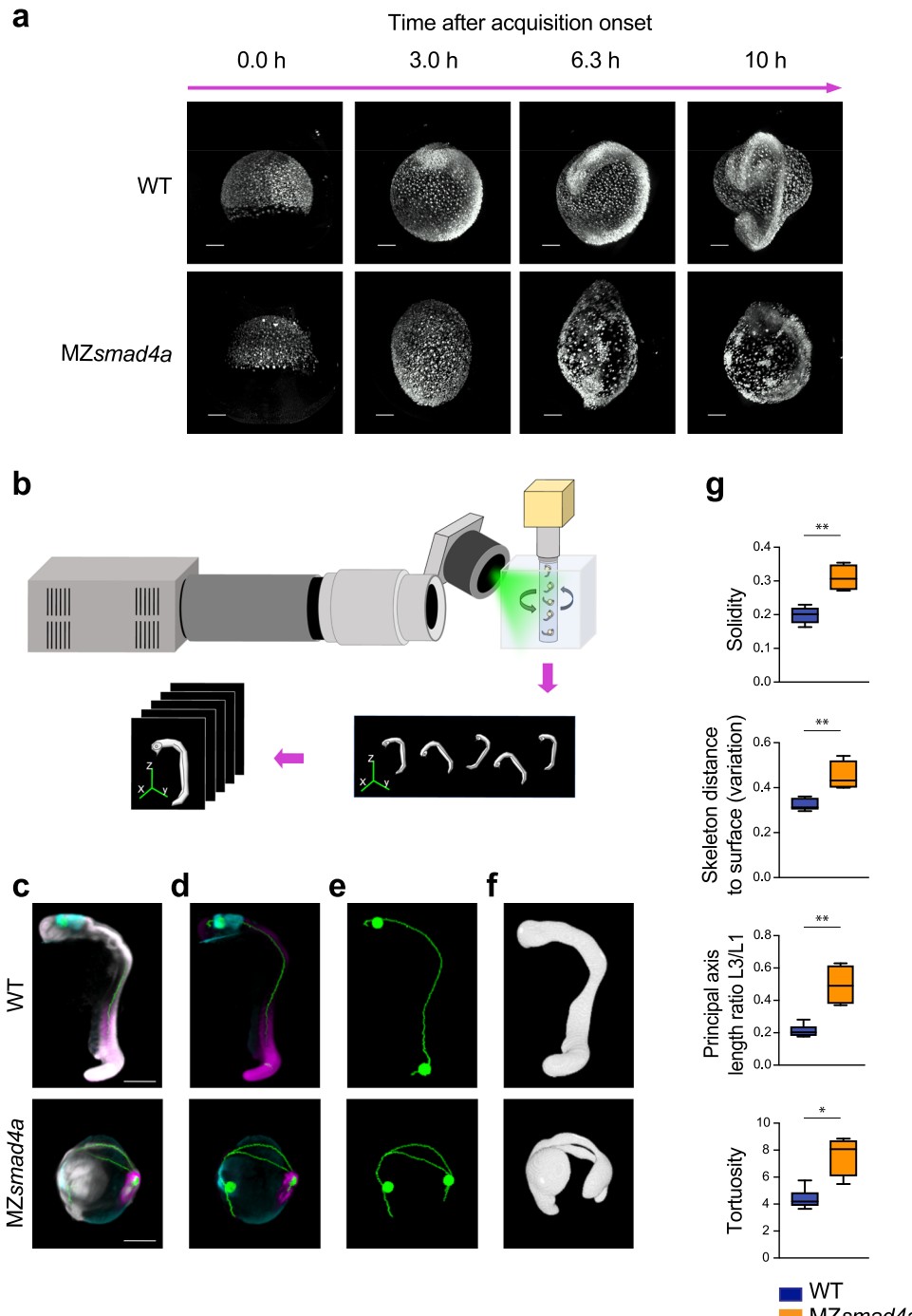

**Fig. 6 An OPT pipeline enabling fast quantitation of 24-hpf zebrafish embryos. a** Snapshots from Supplementary Movies 1 and 2, showing WT and MZ*smad4a* embryo development. Upper and lower panels show developing WT and MZ*smad4a* mutant embryos, respectively. Embryos are shown at 0.0 (mid gastrulation), 3.0, 6.3, and 10 h after acquisition onset. Nuclei were mosaically labeled with PSmOrange. Scale bar corresponds to 150 μm. **b** Schematic of the OPT set up, where up to five embryos can be imaged simultaneously. A 5-channel image is generated for each embryo for subsequent quantitation. **c–f** Examples of processed WT and MZ*smad4a* embryo images. Merged nuclear marker channel together with anterior (*otx2*) and posterior (*myod*) markers and digital skeleton (**c**). Merged anterior (*otx2*) and posterior (*myod*) markers and skeleton (**d**). Skeleton channel, together with anterior/posterior landmarks (**e**). Green spheres along the skeleton mark A/P coordinates. Segmented embryo mask representative of the nuclear marker (**f**). Scale bars correspond to 260 μm. **g** A subset of morphological descriptors from the output array (4 out of 26). Note that all measures, beside the A/P index, refer to the segmented embryo mask. Box-and-whiskers plots show WT ($n = 6$) vs MZ*smad4a* mutants ($n = 4$). Box at 25–75$^{th}$ percentile, whiskers at minimum and maximum values, line at median. Solidity: $p = 0.009$, Skeleton distance to surface (variation): $p = 0.009$, Tortuosity $p = 0.019$, Principal axis length ratio L3/L1: $p = 0.009$. Two sided Mann–Whitney test. *$p < 0.05$; **$p < 0.01$.

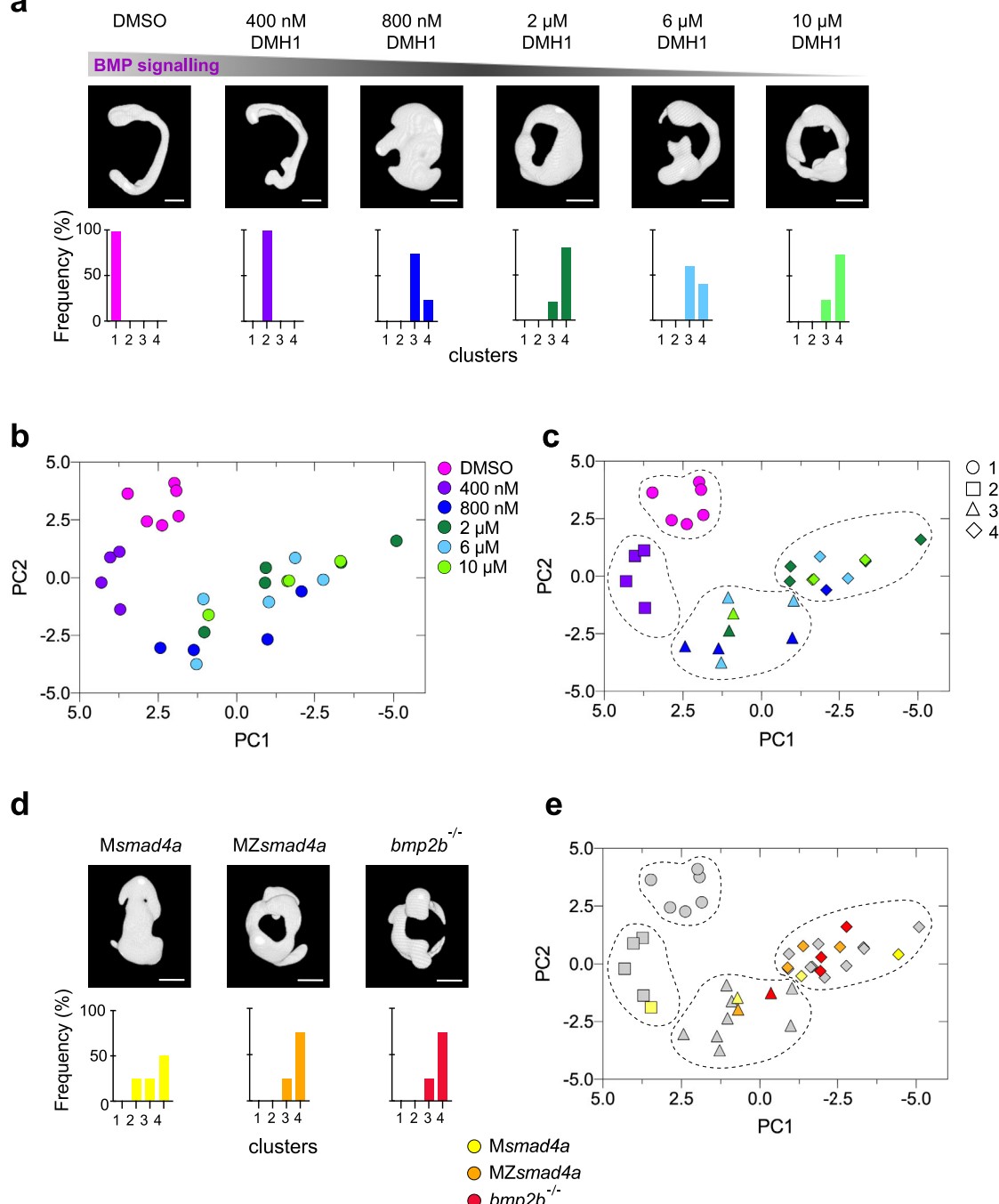

**Fig. 7 Building a morphometric map for BMP signaling. a** Representative embryos treated with different doses of the BMP inhibitor DMH1 are shown. Bottom panels show the percentage of embryos falling in the different severity clusters defined in **c**. Scale bars correspond to 260 μm. **b** Morphometric map for the BMP dose/response. Each embryo is represented as a dot and color coding indicates the DMH1 dose. **c** As in **b**, but using hierarchical clustering, which defines four different clusters with different severity. Color coding indicates DMH1 dose; shapes indicate severity groups. **d** As in **a** but shows embryo masks for MZsmad4a, Msmad4a, and bmp2b[-/-] embryos. Scale bars correspond to 260 μm. **e** MZsmad4a (orange), Msmad4a (yellow), and bmp2b[-/-] (red) are plotted onto the dose/response morphometric map (gray). Shapes indicate severity group.

are similar between WT and MZsmad4a mutant embryos over a time course from 256 cells to 75% epiboly, a process likely influenced by the initial ndr1/2 induction in the YSL. In contrast, expression of lft1/2 over the same time course is initially reduced in the MZsmad4a mutant embryos, before reaching WT levels by 50% epiboly. The differential effect of loss of Smad4 on the expression of ligands and antagonists, which function as a feedback loop, results in a compensation mechanism whereby Nodal signaling (as read out by pSmad2 levels) is prematurely enhanced at early gastrulation, and

then curtailed, as levels of Lft1/2 are restored to WT. We show that this Smad4-independent Nodal signaling is more sensitive to pathway perturbations. We therefore conclude that Smad4, by promoting efficient Nodal-induced transcription, confers robustness. Although Smad4-independent Nodal signaling is sufficient for the specification of axial and paraxial mesodermal gene programs, endoderm induction is clearly impaired. Genetic titration of Nodal signaling levels in mouse embryos revealed that while induction of intermediate mesodermal derivatives like the node and notochord

are induced at low Nodal signaling levels, definitive endoderm and prechordal plate induction require stronger signaling[25,26]. Therefore, it is also likely in MZ*smad4a* mutants that Nodal signaling does not reach the threshold for endoderm induction. Taken together, our findings suggest that while Smad4 is generally dispensable for Nodal signaling, it is important in contexts where high Nodal signaling levels are required for cell fate specification.

**OPT enables the quantification of morphogenetic features in large numbers of whole zebrafish embryos**. Phenotypic analysis of zebrafish embryos has widely served as a gateway to identify new genes and determine their function. This has been achieved either by investigating mutations that co-segregate with developmental phenotypes (forward genetics) or by interrogating the functional consequences of known genetic manipulations (reverse genetics)[90]. In the context of BMP signaling, visual characterization and annotation of phenotypic traits via "dorsalization" or "ventralization" scales have been particularly useful for establishing epistatic relationships between components of the BMP signaling pathway, as well as the phenotypic consequences of targeted pathway manipulations[37,39,45]. However, these approaches are not able to quantitate phenotypic differences across different mutants, especially when the mutant phenotype is the result of complex signaling and genetic interactions. To overcome these drawbacks, we have developed a standardized method for quantitating zebrafish embryo morphology.

We have taken advantage of OPT, which enables fast acquisition of whole 24-hpf embryos without the image burden of light-sheet microscopy, or the in-depth imaging limitations of confocal microscopy[81,91]. By combining this with semi-automated segmentation and quantitation we have been able to visualize individual embryos within a morphogenetic map and explore their relationships. We have captured the morphological consequences of different degrees of BMP inhibition and have generated a BMP morphospace in which embryos are organized into distinct clusters. We have tested if and to what extent Smad4a mutant morphology depends on the loss of BMP signaling and found that MZ*smad4a* and M*smad4a* embryos fall in different positions within these defined clusters in accordance with their morphological features. Additionally, analysis of the BMP morphospace revealed that embryos treated with higher doses of the BMP signaling inhibitor, DMH1, as well as mutant embryos, exhibited more phenotypic variability than did WT embryos, a phenomenon known as "decanalization"[92]. By generating different phenotypic spaces, through the use of different mutants and inhibitors, our approach could be extended and integrated to other signaling pathways apart from Nodal and BMP. Finally, our quantitative approach to defining morphological embryo features opens up a method to combine morphological data with genomic datasets, for example, RNA-seq or ChIP/ATAC-seq, within multidimensional variable space. This will facilitate genotype–phenotype correlations regardless of whether the starting point of the analysis is a gene or a phenotype. We can foresee exciting applications for our approach, not only in basic research, but also for drug screening and drug discovery.

## Methods

**Fish lines and maintenance**. *Danio rerio* were housed in 28.0 °C water (pH 7.5 and conductivity 500 μS) with a 15 h on/9 h off light cycle. All zebrafish husbandry was performed under standard conditions in accordance with institutional (Francis Crick Institute) and national (UK) ethical and animal welfare regulations. All regulated procedures were carried out at the Francis Crick Institute in accordance with UK Home Office regulations under project license P83B37B3C, which underwent full ethical review and approval by the Francis Crick Institute's Animal Ethics Committee.

**Smad4a mutant generation**. *smad4* is found in two copies within the zebrafish genome: *smad4a* (ENSDARG00000075226) and *smad4*b

(ENSDARG00000012649). Note that in the latest release of the zebrafish genome (GRCz11), *smad4b* is composed of two separate transcripts. In the previous release (GRCz10), it is clearly one transcript encoding a protein highly homologous to Smad4a. One-cell stage zebrafish embryos were injected with 300 ng/μl of Cas9 mRNA[93] and 50 ng/μl of *smad4a* gRNA in injection buffer (10x stock: 200 mM HEPES pH7.0, 1.2 M KCl) as described[93]. Injected embryos were grown to adulthood and outcrossed with WT. Adult progeny were screened via PCR and Sanger sequencing using the primers listed in Supplementary Table 1. F0 embryos were grown and fish carrying the deletion were selected as founders. For routine breeding and maintenance of the mutant strain, genotyping was outsourced to "Transnetyx [https://www.transnetyx.com/]".

**OPT**. Zebrafish embryos were imaged using the 4x imaging arm of a dual-magnification OPT system that also has a 1x arm. The 4x arm used comprises a microscope objective lens (Plan Fluor 4x/0.13, Nikon) and tube lens (TTL-200A, Thorlabs), with an adjustable aperture placed behind the microscope objective lens to enable the system numerical aperture to be adjusted in order to maintain object-space telecentricity. Illumination was provided by a multiline laser light source (TriLine, Cairn Research Ltd), which has 460, 555, and 635 nm lines available, that is coupled through an actively vibrated 800 μm core diameter fiber in order to homogenize the beam profile. The beam from the fiber output was collimated, passed through a motorized excitation filter wheel, then focused onto a diffuser for further homogenization, after which a Fresnel lens (FRP251, Thorlabs) was used to collimate the light coming from the diffuser and to illuminate the sample. When imaging smaller samples, this Fresnel lens can be moved further from the diffuser to partially focus the illumination on a smaller sample region to increase the illumination flux if desired. The fluorescence images were captured by an sCMOS camera (Zyla 5.5, Andor) in front of which was an emission filter. The EGFP/Fluorescein-TSA excitation (FF01-445/45-25) and emission filters (FF01-535/50-25), TdT/SYTOX Orange excitation (FF01-554/23-25) and emission filters (FF01-609/54-25), and the AF647/Fast Blue excitation (FF01-631/36-25) and emission filters (FF01-708/75-25) used were all from Semrock. OPT fluorescence image data were acquired at 400 equally spaced projection angles, with the integration time of each project image being on the order of 25–500 ms, depending on the fluorophore.

**OPT reconstruction and segmentation**. The OPT projection images acquired in different fluorescence channels were first corrected for the mismatch of the rotation axis with the middle line of camera's field of view. Briefly, the number ($n = 16$) of pairs of opposite-side projections were first transformed to the Gaussian gradient modulus with $\sigma = 30\,\mu m$ smoothing using the "MathWorks Nonlinear Diffusion Toolbox [https://www.mathworks.com/matlabcentral/fileexchange/3710-nonlinear-diffusion-toolbox]" (3 voxels in unbinned data), and then cross-correlated by using the "Matlab *xcorr2* function [https://www.mathworks.com/help/signal/ref/xcorr2.html]" to define rotation axis shifts for every pair. The median value of the calculated shifts was taken, and the projection images were translated correspondingly to correct for this shift. The corrected projections were then reconstructed by using the Filtered Back-Projection algorithm implemented using the "Matlab *iradon* function [https://www.mathworks.com/help/images/ref/iradon.html]". For the purpose of further segmentation and analysis, the reconstructed 3D images of embryos typically containing 5 embryos per acquisition, were rendered in the resolution of 6.5 μm/voxel. Then, the local intensity-based segmentation was applied to the TdT (SYTOX Orange) channel to define the body of the embryo. The other fluorescent images which monitor the expression of gene markers in the head and the tail of the embryo (*otx2*, and *myod*, respectively) were not used for segmentation. The 2-scale sensitive segmentation for SYTOX Orange channel was arranged by using the 3 images $U_{\sigma1}, U_{\sigma2}, U_{\sigma3}$ representing the original SYTOX Orange intensity $U$ smoothed with progressively increasing Gaussian $\sigma$ parameter. These 3 images were used to form the two normalized scale-sensitive difference images as

$$u_1 = (U_{\sigma1} - U_{\sigma2})/U_{\sigma2} \tag{1}$$

$$u_2 = (U_{\sigma2} - U_{\sigma3})/U_{\sigma3} \tag{2}$$

Finally, the weighted scale-sensitive image was synthesized and then thresholded with the help of adjustable weight and threshold parameters $w$ and $t$, as

$$segmentation = w \cdot u_1 + (1 - w) \cdot u_2 > t \tag{3}$$

We found by visual inspection that for the given resolution and quality of images, the values of adjusting parameters providing satisfying segmentations of the dataset, were maintained constant across samples. Out of 43 embryos, segmentation parameters had to be adjusted for one of the 400 nM DMH1 samples due to a yolk artefact (see Supplementary Data File 2 for the specific parameters used). At the last stage of segmentation, each individual binarized embryo body was identified for the analysis by passing the size criterion. Having the geometric positions of the segmented embryo body, we then applied the quantification step of the analysis to characterize every embryo by a feature vector of morphological parameters. In particular, we calculated the shape and intensity-based descriptors given in Supplementary Data File 2. Intensity-based quantifiers included the correlation of the distance maps seeded by the anterior and posterior end of an embryo identified as maximum brightness points in the corresponding *otx2* (anterior), and *myod* (posterior) spectral channels.

To get statistics on different sets of voxels, we calculated the variation (standard deviation over mean), and the skewness and the kurtosis of the distribution. For image intensity, the sample minimum was subtracted. The code for the "segmentation [https://github.com/yalexand/ALYtools/blob/master/Classes/%40ALYtools_data_controller/do_OPT_ZFish_Embryo_Segmentation.m]" and "quantitation [https://github.com/yalexand/ALYtools/blob/master/Classes/%40ALYtools_data_controller/analyze_OPT_ZFish_Embryo.m]" of zebrafish embryo shape was implemented in Matlab (Mathworks)[94]. To study the discrimination ability of the parameters, we considered the embryos in the multidimensional feature space. For this, we used the "Factoextra [https://rpkgs.datanovia.com/factoextra/index.html]" package in R. Briefly, the selected morphological descriptors were scaled and visualized using PCA-based dimensionality reduction[95] using the prcomp function. Next the contribution of the different principal components to the total variance was visualized using the fviz_eig function. As PC1 and PC2 contributed most to the variance, they were chosen as the main PCA axes. To better interpret the morphogenetic map, we visualized the contribution of the different variables for the first two components using the Fviz_contrib function. To visualize the PCA space, PC1 and PC2 coordinates were extracted and used for plotting in Prism9. The Factoextra package was also used to group embryos using default clustering algorithms. Despite the small number of data points, unsupervised methods such as hierarchical clustering[96] K-means[97], or K-medoids[98] provided similar partition patterns into four clusters, roughly reflecting their observed actual morphological differences ('unperturbed','slightly','moderately' and 'heavily perturbed'). Finally, rendering of OPT image data was performed with the software Icy (Institute Pasteur and France bioimaging) and Imaris (Oxford instruments) for representative figures and movies. Renderings were adjusted to enhance contrast where appropriate.

**Single cell isolation and sequencing**. Fertilized embryos were kept in embryo media until they reached high stage. At that time, they were transferred to agar-coated 6-cm petri dishes and manually dechorionated in embryo media. At sphere stage, embryo media was sequentially replaced with Dulbecco Modified Eagle's Medium (DMEM F-12, ThermoFisher Scientific) containing 0.04% BSA, and the blastula cups were dissected away from 25 embryos and transferred to an eppendorf containing 100 µl of DMEM-F12 + 0.04% BSA. Cells were then pelleted at 400 × g for 30 s. The media was removed, and cells were suspended in 200 µl of ice-cold phosphate buffered saline (PBS) containing 0.04% BSA, to which 800 µl methanol were added dropwise. Samples were kept on ice for 15 min and finally transferred to −20 °C until use. For rehydration, samples were incubated on ice for 15 min and then cells were pelleted at 1000 × g for 5 min at 4 °C. Cells were resuspended in 100 µl ice cold PBS containing 0.04% BSA and the cell suspension was loaded into 10X Genomics Single Cell 3′ Chip using a Chromium 10X machine (10X). The library was sequenced with an Illumina HiSeq 4000.

**ScRNA-seq analysis**. For the sphere sc-RNAseq library, quantification and alignment were carried out using cellranger-2.1.1 against genome GRCz10-release-89 from Ensembl. Initially, the dataset was analyzed using the cell ranger tool (10X) which estimated 2644 sequenced cells, with a total of 157,991,051 reads, 59,754 mean reads per cell and 3807 median genes per cell. The sequencing saturation was 62%. Further analysis of the dataset was performed using the "Seurat R toolkit [https://satijalab.org/seurat/articles/get_started.html]"[99]. First, cells with high mitochondrial counts were removed from the dataset using a 10% threshold filter which removed 49 cells from the dataset. Next, counts were normalized, and highly variable features were selected and scaled using default parameters. After PCA, cells were classified in clusters (find cluster resolution = 0.2) and displayed as uMAPs or heatmaps. For uMAPs and clustering the first 15 PCs were considered. Expression of different gene features is displayed on uMAPs in the form of normalized counts. The different clusters were characterized based on the markers shown in Supplementary Fig. 2a. Of the 4 different clusters, cluster 4 was more difficult to characterize due to the low number of cells. This cluster did not present particularly high levels of mitochondrial counts but shared some markers with the EVL and displayed expression of YSL markers. As the YSL forms from the regression of pre-existing cell membrane from marginal blastomeres[100], we hypothesized these could be YSL progenitors and named them putative YSL.

**RNA extraction**. For RNA extraction of embryos, 15 (for qPCR) or 25 (Bulk RNA-seq) embryos were collected for each biological replicate. Embryos were homogenized in Trizol (ThermoFisher Scientific) using a 1 ml syringe (BD Plastipack) equipped with a 21 G x 1/2 needle (BD Microlance). Water-saturated chloroform (200 µl) was then added to the samples and mixed by vortexing. The samples were centrifuged for 15 min at 16,200 × g at 4 °C and the aqueous phase was added to 500 µl isopropanol to precipitate the RNA. Total RNA was pelleted by centrifugation at 16,200 × g for 45 min at 4 °C. The pellet was then washed with ethanol, centrifuged again, and resuspended in 50 µl of RNAse-free water. Residual DNA in the samples was eliminated by incubating the samples in DNAse (Worthington Biochemicals) at 37 °C for 10 min. Total RNA was purified from the DNAse reaction by adding 150 µl of phenol-chloroform isoamyl alcohol (Invitrogen) followed by centrifugation at 16,200 × g for 5 min. The aqueous phase was mixed with 300 µl of 100% ethanol and 25 µl of ammonium acetate (4 M pH 5.6) to precipitate the RNA. Finally, isolated RNA was pelleted by centrifugation for

15 min at 16,200 × g at 4 °C and the pellets were washed in ethanol and re-suspended in 30 µl of RNAse-free water. RNA extraction of mouse ESCs was performed as previously described[101].

**Bulk RNA-seq**. For the bulk RNA-seq experiment, three biological replicates were used. The quality of the extracted RNA was assessed using a bioanalyzer (Agilent). Libraries were prepared using the KAPA mRNA HyperPrep kit (Roche) and paired-end reads were generated using an Illumina HiSeq 4000.

**RNA-seq quantification and differential expression analysis**. Biological replicate datasets were analyzed using the Francis Crick Institute "BABS-RNASeq Nextflow pipeline [https://github.com/crickbabs/BABS-RNASeq]"[102]. The GRCz11 zebrafish reference genome was used with the Ensembl release-95[103] gene annotations. The BABS-RNASeq pipeline implements steps to assess dataset quality and quantifies gene expression. Illumina adapter contamination was removed from reads using cutadapt[104]. Dataset quality and replication was validated using "FastQC [https://www.bioinformatics.babraham.ac.uk/projects/fastqc/]", RSeQC[105], RNA-SeQC[106], and "Picard [http://broadinstitute.github.io/picard/]". Reads were then aligned to the genome and expression quantified using STAR[107] and RSEM[108]. Estimates of gene expression were loaded into R[109] and the expected counts converted to integers. The counts matrix was normalized and differentially expressed genes were identified using DESeq2[110] where FDR ≤ 0.01.

**cDNA preparation and qPCR**. For both mESCs and for zebrafish embryos cDNA was generated with the AffinityScript kit (Agilent) as previously described[34]. Briefly, 500 ng of RNA were retrotranscribed using random primers. For qPCR, the cDNA was diluted 1:10. All qPCRs were performed with the PowerUp SYBR Green Master Mix (ThermoFisher Scientific) with 300 nM of each primer and 2 µl of diluted cDNA. Fluorescence acquisition was performed on either a QuantStudio 5 System machine or QuantStudio 12 Flex (ThermoFisher Scientific). Primers are listed in Supplementary Table 1. For the zebrafish samples, quantification for relative gene expression was performed using the comparative Ct method and target gene expression was normalized to *actin*. In addition, for most of the qPCRs, time course values for MZ*smad4a* and M*smad4a* mutants were normalized to the WT expression levels at 75% epiboly for each replicate. For SB-505124 dose responses, WT and MZ*smad4a* mutant values were normalized to each DMSO treatment value. Instead, for mESCs experiments, target gene expression was normalized to *Gapdh* levels. Fold changes in gene expression between WT cells and SMAD4 knockout clones were obtained by normalizing expression values to the 2-h WT sample in the case of the BMP4 inductions and to the 8-h WT sample in the case of the Activin A inductions.

**Single and double-FISH**. The majority of the probes were transcribed from linearized plasmids as previously described[34]. Details of the plasmids, with the relevant restriction enzyme and polymerases are given in Supplementary Table 2. The *tbx16* probe was transcribed from a PCR fragment, which was generated via PCR from cDNA using the primers listed in Supplementary Table 1. PCR was performed using the Phusion high fidelity DNA polymerase (NEB).

FISH was performed as previously described[34]. Briefly, embryos were incubated in 2% H₂O₂ in 100% methanol for 20 min to clear background staining before rehydration with PBS/0.1% Tween 20 (PTW). For single FISH, embryos were incubated first with a Digoxigenin (Dig)-11-UTP-(Roche) labeled probe at 65 °C overnight and, after extensive washing in SSC buffer, with an anti-Dig-HRP (Roche, 1:500). For double-FISH, the Dig-probe was incubated in combination with a dinitrophenol (DNP)-11-UTP (Perkin Elmer) labeled probe, followed by incubation with an anti-DIG-HRP plus an anti-DNP-AP (Vector labs MB-3100, 1:1000) antibody. To detect HRP, embryos were first pre-incubated with tyramide (Sigma) coupled with fluorescein-NHS ester (ThermoFisher Scientific) for 25 min in the dark in PTW. Next 0.001% H₂O₂ was added, and embryos were incubated for an additional 30 min to allow the signal to develop. For double-FISH, after the HRP reaction, embryos were washed several times and incubated with a Fast Blue + Naphthol-AS-MX solution (Sigma). This allowed for alkaline phosphatase detection. Finally, nuclear staining was performed with either DAPI 1:1000 in PTW or SYTOX Orange (Invitrogen) at 1:200,000 in PTW at 4 °C overnight.

**Quantitative in-situ hybridization (RNAscope®)**. Quantitative FISH was performed with the RNAscope® 2.0 Assay[111] using the Multiplex Fluorescent Assay v2 (ACDBio) according to manufacturer's instruction. Briefly, after fixation and rehydration embryos were incubated with Dr-*lft1* (557771-C2, ACDBio) or Dr-*ndr1* (557761, ACDBio) probes at 40 °C overnight. Embryos were then washed 3 × 15 min in 0.2x saline sodium citrate/0.01% Tween 20 (SSCT) and re-fixed in 4% PFA at RT for 10 min. After washing, embryos underwent a series of incubations interspersed with 4 x 5 min washes with 0.2 x SSCT. First, they were incubated with two drops of the Amp1 and Amp2 solution at 40 °C for 30 min and then incubated with two drops of Amp3 at 40 °C for 15 min. After an additional washing step, embryos were incubated with two drops of the Multiplex FL V2 HRP-C1 (*ndr1*) or C2 (*lft1*) at 40 °C for 15 min. After a last series of washes in SSCT, embryos were washed in PTW and processed for the staining. Like conventional FISH, embryos were incubated with tyramide (Sigma) coupled with fluorescein-NHS ester

(ThermoFisher Scientific) in PTW in the dark. To allow HRP detection, 0.001% $H_2O_2$ was added to the reaction and embryos were incubated for 30 min, also in the dark. The embryos were then extensively washed and DAPI was used at 1:1000 in PTW overnight at 4 °C.

**Immunostaining.** For immunostaining, after rehydration, embryos were washed extensively in PBS/1% Triton X-100 and incubated in cold acetone at −20 °C for 20 min[33]. They were then blocked in 10% FBS and 0.8% Triton X-100 plus 1% DMSO in PBS. Embryos were then incubated with either primary antibodies against pSmad2 (Cat# 8828, Cell Signaling Technology; 1:800) or pSmad1/5 (Cat# 13820, Cell Signaling Technology; 1:800) at 4 °C overnight. Antibody binding was detected with a donkey anti-rabbit Alexa Fluor 488 (Cat# A-21206, ThermoFisher Scientific, 1:400). For nuclear staining embryos were incubating with DAPI 1:1000 in PTW at 4 °C overnight.

**Confocal imaging and image analysis.** Confocal images (Fig. 1d–g, Supplementary Figs. 1c–f and 5c) were acquired with an inverted Leica Sp8 laser scanning confocal microscope (Leica Biosystems). Embryos were mounted in 35 mm Mat-Tek dishes (MatTek) in 1% low melting agar (Sigma) and Z-projections were acquired with a 10X dry lens. Alternatively, (Figs. 3b, d, 4a, d, e, h and 5d, Supplementary Figs 3c, 5a, b, f, h, 6c) embryos were imaged with an upright Leica Sp5 (Leica Biosystems) with a 20X HCX PL APO lens (Leica Biosystems) and embryos were mounted in custom dishes in 1% low melting agar. For quantification of the pSmad1/5 and *chrd* expression domains the Polygon selection tool (Fiji) and the straight-line tool (Fiji) were used on MIPs to define a ROI and measure the areas and the lengths of the expression domains. For the quantification of signaling levels and gene expression at the margin, embryos were mounted with the lateral margin perpendicular to the lens, and confocal stacks (about 200 μm) were acquired with a 2 μm Z-interval. Using the ImageJ software[112], a region of interest (ROI) was drawn on each optical section to exclude nuclei from the YSL. Next, individual nuclei were segmented from the DAPI staining from each optical section by thresholding with a mean filter. Binary images were refined by an erode and dilate step (Fiji binary functions) and the resulting mask was used to measure nuclear intensities for pSmad2, *ndr1* and *lft1* with the 3D object counter function. To obtain the position for each nucleus with respect to the margin the coordinates of the centroids for segmented nuclei were recorded across the *y* axis. The resulting nuclear intensities for the different markers were visualized either by plotting individual nuclei values or alternatively, for each embryo, nuclear intensities were fitted with a Lowless function using the software PRISM8 (Graphpad) and curves from mutant and control groups were merged and visualized as mean and SEM. For illustrative purposes, maximum intensity Z-projections from confocal stacks were adjusted to enhance brightness where appropriate. Adjustments were kept equal between control and treated or mutant samples for all images, apart from in Fig. 1f where the DAPI signal was slightly enhanced with respect to Fig. 1d, e, g.

**Light sheet microscopy.** Light sheet imaging was performed using a Luxendo MuVi-SPIM microscope, PSmOrange labeling was detected by using a 515 nm excitation laser and a LP530 filter. Each embryo was acquired from mid-gastrulation every 5 min for 11–12 h in line mode configuration. Movies were processed using the Imaris software.

**Plasmids and RNA synthesis.** The zebrafish open reading frames (ORF) for *smad4a* and *smad4b* were PCR amplified with oligonucleotides containing BamH1 and XhoI sites at their 5′ and 3′ ends for *smad4b* and ClaI and XhoI sites at their 5′ and 3′ ends for *smad4a* and were ligated into the pCS2 vector. Similarly, human *BMP4* was cloned into pCS2 using BamHI and XhoI. The gRNA plasmid for *smad4a* was generated by annealing gRNA oligos and subsequent ligation into the pT7-gRNA vector[93]. Details of all plasmids used are given in Supplementary Table 2. Primer and oligonucleotide sequences are given in Supplementary Table 1. Capped RNA for injection was transcribed as previously described[34] except for PSmOrange RNA which was transcribed using the mMessage mMachine Sp6 kit (ThermoFisher Scientific) followed by LiCl precipitation. gRNA for *smad4a* was transcribed using the T7 MEGAscript Kit (ThermoFisher Scientific), followed by phenol/chloroform extraction and isopropanol precipitation.

**Morpholino and mRNA injections.** For rescue experiments 80 pg of *SMAD4* mRNA or *smad4a* mRNA was injected at the one-cell stages. *BMP4* mRNA was injected at either 60 pg or 20 pg. The MO for *smad4b* or a mispaired version (Genetools; see Supplementary Table 2 for the sequences) were diluted in $H_2O$ to a final working concentration of 1 mM. For each MO, 2 pmol were injected in 1–2 cell stage embryos in a volume of 2 nl. Considering that both qPCR and bulk-RNAseq experiments showed absent expression for *smad4b* during early development, the MO's efficacy was assessed by injection of 80 pg of *smad4b* mRNA into MZ*smad4a* mutants with or without the *smad4b* MO or the *smad4b* mpMO.

**Chemical inhibition.** The inhibitors DMH1 (Selleck Chemicals) and SB-505124 (Sigma) were dissolved in DMSO and directly diluted in embryo medium. Drug

dilutions are indicated in the text and figure legends. For the dose-responses, concentrations ranged between 10 μM and 400 nM for both drugs.

**Cell culture.** HaCaT cells were obtained from the Francis Crick Institute Cell Services from stocks deposited in the ICRF Cell Bank by Lional Crawford in 1994. HaCaT SMAD4 knockout clones were generated at the Francis Crick Institute using CRISPR/Cas9 technology[113]. All three cell lines were banked by the Francis Crick Institute Cell Services, and certified negative for mycoplasma. Cells were maintained in Dulbecco's modified Eagle's medium (DMEM) supplemented with 10% FBS and 1% Penicillin/Streptomycin at 37 °C with 10% $CO_2$. The feeder-independent E14-TG2a mouse embryonic stem cells carrying a *Hhex*-RedStar/*Gsc*-GFP reporter[114] were obtained from Joshua M. Brickman (DanStem, University of Copenhagen) and were also certified mycoplasma negative by the Francis Crick Institute Cell Services. Cells were maintained in Glasgow Minimal Essential Medium (GMEM, Gibco), containing 10% FBS (Gibco, batch tested), 2 mM L-glutamine (Life Technologies), 1 mM sodium pyruvate (Life Technologies), 0.1 mM non-essential amino acids (Life Technologies), and 0.1 mM 2-mercaptoethanol (Sigma). 1000 U/ml leukemia inhibitory factor (LIF) was added just before use. Prior to the cell seeding, culture flasks and plates were coated with 0.1% gelatine (Sigma) in PBS. The cells were incubated at 37 °C with 5% $CO_2$.

**Generation of SMAD4 knockout mESCs.** Guide RNAs targeting the MH1 domain of SMAD4 (see Supplementary Table 1) were cloned into the pSpCas9(BB)-2A-GFP (PX458) plasmid[113]. 700,000 E14-TG2a mESCs were seeded in a 10-cm dish in a total volume of 10 ml GMEM media. The next day, the transfection mix consisting of 7.5 μg of pX458-gRNA plasmid, 60 μl of FuGENE HD (Promega) in a total of 1 ml Opti-MEM™ (ThermoFisher Scientific) was prepared according to the FuGENE HD (Promega) protocol and then added to the cells. Twenty-four hours post transfection, E14-TG2a cells were bulk sorted for GFP expression and then sparsely plated in 15-cm dishes to form single cell colonies. Ten days post-sorting, 30 clones were picked and transferred into 96-well plates. Once confluent, SMAD4 protein deletion was verified by Western blotting and selected clones were further validated by Sanger sequencing. For this, cells were lysed with the QuickExtract DNA Extraction Solution (Lucigen) according to the manufacturer's instructions. PCR products were generated using primers targeting upstream and downstream of the CRISPR/Cas9-targeted sequence (Supplementary Table 1) were ligated into pGEM-T vectors and after bacterial transformation, clones were picked for plasmid extraction and Sanger sequencing.

**Ligands and cell treatments.** Activin A (PeproTech, 120-14E) was reconstituted in PBS supplemented with 0.1% BSA. BMP4 (PeproTech, 120-05ET) was reconstituted in 4 mM HCl supplemented with 0.1% BSA. Ligands were used at 20 ng/ml. Cells were treated with the indicated ligands for the times indicated in the figures. SB-431542 (Tocris, UK) was used at a final concentration of 10 μM. Prior to Activin A stimulation cells were starved with 10 μM SB-431542 overnight and then the SB-431542 was washed out prior to Activin A stimulation. For BMP4 treatments, cells were serum-starved overnight and then treated with BMP4.

**DNA pulldown assay and Western blotting.** Parental HaCaT and SMAD4 knockout clones were serum starved overnight with Opti-MEM™ (ThermoFisher Scientific) and then were treated with 20 ng/ml BMP4 (PeproTech) for 1 h. Nuclear lysates were prepared using buffer containing 420 mM NaCl as described[115]. DNA pulldown assays were performed as previously described with some modifications[116]. DNA pulldowns were performed in the presence of a 20 μg of non-biotinylated mutant competitor oligonucleotide to reduce non-specific binding. The oligonucleotides corresponding to WT and mutated upstream *ID3* enhancer[62] are shown in Supplementary Table 1. Western blotting was carried out using standard methods. The list of the antibodies used is shown in Supplementary Table 2, as are the dilutions at which they were used.

**Statistical analysis.** All qPCR data are the means and SEMs of at least three independent biological experiments, except the gene expression for *smad4a* and *smad4b* after gastrulation which came from two independent experiments. Within each qPCR experiment technical duplicates were run. All immunofluorescence and in-situ experiments were performed at least twice and the majority, at least three times. qPCR time series experiments were analyzed using Microsoft Excel and statistical analysis was performed using the software Prism8 with an unpaired multiple comparison *t*-test with Holm-Sidack correction. For comparisons between more than two groups, 2-way Anova and Tukey post hoc test were used. For immunofluorescence and OPT data on individual descriptors a non-parametric Mann–Whitney test was used. For the bulk RNA-seq quantification of gene expression differences between WT and MZ*smad4a* replicates a Wald test was used. *$p < 0.05$, **$p < 0.01$, ***$p < 0.001$, ****$p < 0.0001$.

**Reporting summary**. Further information on research design is available in the Nature Research Reporting Summary linked to this article.

## Data availability

The bulk RNA-seq and scRNA-seq data have been deposited in GEO under the accession code "GSE162289" for the bulk RNA-seq and "GSE164574" for the scRNA-seq. The sc-RNA seq analysis is accessible as an "R markdown file" (Supplementary Data File 3). All quantitative data presented in the paper are supplied in the Source Data file, as are the uncropped Western blots shown in Fig. 3 and Supplementary Fig. 4. All other relevant data supporting the key findings of this study are available within the article and its Supplementary Information files or from the corresponding author upon reasonable request. Source data are provided with this paper.

## Code availabilty

The custom code used in this study for the segmentation and quantitation of zebrafish embryo shape is available on "Github [https://github.com/yalexand/ALYtools]"[94]. Source data are provided with this paper.

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

## Acknowledgements

We thank Mary Mullins and Steve Wilson for zebrafish lines, Josh Brickman for the mouse ES cell line, and Matthias Carl for reagents. We are very grateful to Mollie Millington and the Francis Crick Institute Aquarium team, in particular Sarah Wheatley, to the Advanced Sequencing Facility, in particular, Amelia Edwards, to the Light Microscopy Facility, Cell Services and the Genomics Equipment Park. We thank Alessandro Ciccarelli for invaluable help with the light sheet imaging. We thank James Briscoe, Matthias Carl, Davide Coda, Nic Tapon, and all the members of the Hill lab for helpful discussions and very useful comments on the manuscript. For the purpose of Open Access, the corresponding author has applied a CC BY public copyright licence to any Author Accepted Manuscript version arising from this submission. This work was supported by the Francis Crick Institute which receives its core funding from Cancer Research UK (FC001095), the UK Medical Research Council (FC001095), and the Wellcome Trust (FC001095).

## Author contributions

L.G., C.H., and C.S.H. conceived and designed the study. L.G., C.H., S.K., I.G., and F.P. planned, performed the experiments, and analyzed the data, with help from A.D.E. Additional data analysis was performed by Y.A., C.B., and P.E. P.M.W.F and J.M. provided resources and supervision. L.G. and C.S.H wrote the manuscript. C.S.H. provided supervision and funding for the study.

## Competing interests

The authors declare no competing interests.

## Additional information

**Peer review information** *Nature Communications* thanks Danny Huylebroeck and the other anonymous reviewer(s) for their contribution to the peer review this work. Peer reviewer reports are available.

