## [Peer Review File · Nature Communications]

Reviewers' Comments:

Reviewer #1:

Remarks to the Author:

Through a sophisticated combination of careful smad4a (and 4b, which does not need further attention and is also not a compensator of smad4a deficiency) gene expression studies in zebrafish early embryos (main figures) and larvae (suppl. figs), CRISPR/Cas based knockout, transcriptomics, live imaging including assessing embryo shape, and also using light sheet microscopy and OPT (the latter having innovative merit on its own, also because of the incorporated morphometry), Gulielmi and co-workers of the team of Caroline Hill present a compelling case for a distinct Smad4a effector dependency for BMP and Nodal signaling and proper regulation of the respective target genes in the developing embryo. It leads to a number of important observations and conclusions of developmental signaling by TGFbeta family subgroup ligands. It is also going to be relevant for congenital and chronic disease, including cancer. Indeed, this work is entirely novel and its results are extremely important for the TGFbeta family signaling field, for it is a significant step beyond previous and more superficial, predominantly cell culture work or studies in worm, fly and vertebrate embryos. This is altogether a superb study about which only minor remarks apply. It will fit perfectly the top-quality requirements by your journal. The minor remarks are:

- The authors are perfectly aware that certainly and par excellence BMP, but also Nodal signaling in the embryo is controlled by autoregulation, synexpression and feedback mechanisms, including on shared or separate components of these two systems, many of which they investigate, including by RNA-seq of MZsmad4a embryos at 50% epiboly (the appropriately selected stage). The already impressive results of this approach could however be discussed more in detail. Assuming they have the RNA-seq results and the data are there already, it would for example be interesting to discuss on top of the downregulated BMP targets within the batch of changed genes (i) also the upregulated genes in the KOs (any of these also known – and biochemically of functionally proven – as modulators of BMP/Nodal signaling?), and report whether (ii) levels of TIF1gamma/Trim33 change (the ratio of this repressor or alternative transcription factor that promotes TGFbeta signaling to Smad4 is important, in particular for short stimulation times that mediate higher threshold responses) and (iii) levels of TTF-1/Nkx2.1 might have changed. Although in particular the latter, for reasons of temporal and domain expression, may be unlikely, it is recommended to have a look at this and perhaps even mention explicitly, as this is a Smad4 complex disruptor (albeit best documented in Smad3-dependent EMT) prohibiting Smad4 binding to chromatin and hence is part of Smad4-independent regulation.
- In the section addressing the requirement of smad4 for P-Smad1/5 to bind to DNA, reference (i.e. refs. 55-57) is made to binding motifs. However, here it would be appropriate to also cite the identification of CG-rich sequences by Smad1/5 by e.g. the Niehrs group, but certainly also of Morikawa and Miyazono teams (who used BMP-Smad ChIP-seq). The latter teams identified the "GC" motif (high-affinity, so low BMP doses are sufficient) and "GA" motif (low-affinity, so high BMP doses needed) ideally separated by 5 bp from a Smad4 binding site. In this respect, and even taking into account the used dose of 20 ng BMP4/ml – which in the BMP field is an intermediate dose – in the HaCat cells, the analysis of ID3 is relevant, but likely only addresses a "GC" type BMP-Smad/Smad4 motif. The inclusion of a typical "GA" motif containing proximal promoter of another BMP target gene would be recommended here. Also in this system, and similar to one of the aforementioned minor remarks, levels of in particular TIF1gamma/Trim33 could be quickly looked at (and perhaps mentioned) in Smad4 WT versus Smad4 KO cell lines.
- With the experiments that compellingly show that Smad4-independent Nodal signaling is less robust, and where it is speculated that P-Smad2 homotrimers are recruited by transcription factors, it is along similar lines also recommended to look at the level of TTF-1 and, if there, to consider carrying out an extra experiment including TTF-1 morpholinos on top of the other set-ups.

Reviewer #2:

None

Reviewer #3:

Remarks to the Author:

As requested, my review focuses on the technical aspects of the scRNA-seq analysis included in the manuscript.

These analyses were carried out using a pooled sample from 25 embryos. This approach was necessitated by the small numbers of cells that can be obtained from a single embryo and the technical limitations of the scRNA-seq platforms (i.e. fixed scale and limited capture efficiency). However, it can be expected that the amount of inter-individual variability is relatively low during these early stages of development, and therefore the pooling approach does not represent a significant compromise for data interpretation. There appears to be no technical replicates, but as the focus was on analyzing composition of a single sample type (instead of comparisons between samples), in my opinion this is acceptable.

The library prep was carried out using fixed cells, adhering to manufacturer's protocols (10x Genomics). The methods section does not contain all the details that might be relevant to readers, such as read lengths, sequencing depth, number of cells loaded and numbers of genes detected per cell.

Low quality cells were filtered out by setting a threshold value for fraction of reads mapping to mitochondrially-encoded genes. This is a straightforward and commonly used approach, although it is often used in combination with other criteria (e.g. number of UMI or genes detected per cell). The fraction of cells discarded is not specified, although it might be of interest to some readers. The data was analyzed using the Seurat toolkit, which is one of the best established and widely used approaches. The analysis was limited to dimensionality reduction and clustering, representing a straightforward "atlasing" approach. The clustering analysis was performed at relatively low resolution, and the resulting clusters appear well defined (Fig S2A) and correlate with the expression of several known marker genes.

Altogether, in my opinion these analyses are technically sound and the data appears to support the conclusions made by the authors. However, I would encourage the authors to add some details to the methods section.

Reviewer #4:

Remarks to the Author:

In the submitted manuscript, Guglielmi and colleagues tackle a surprisingly unresolved issue in developmental biology: the contribution (or lack thereof) of Smad4 to BMP and Nodal signaling. Smad4 as widely touted "helper Smad" is generally assumed to be an essential co-factor for phosphorylated Smads downstream of the major BMP and Nodal responses in vertebrates - yet, most of this assumption is extrapolated from few *in vivo* validations and mostly from how the TGF-beta pathways work in *Drosophila*. The authors take on this interesting challenge by analyzing the first smad4a mutant in zebrafish, revealing that predominantly BMP signaling is perturbed upon Smad4 loss, while Nodal signaling is still functioning rather well.

The authors introduce their work with an excellent intro section and back up their findings with several classic and new means in an outstanding showcase of technical prowess (and mostly friendly color schemes for color-impaired readers). The findings will be of widespread interest to the field. In its current form, the manuscript will benefit from revisiting the claims and details listed below to strengthen the narrative and context of the presented evidence.

Major Points:

1) The authors go through several iterations of stating that smad4b is unlikely to be of significance to their work and to early Smad4 function in the zebrafish embryo - yet, fact is that smad4b exists, is expressed in early development, and the authors do not provide any genetic evidence for any lack of contribution.

While the authors did not seem to have generated a smad4b mutant (the reviewer does hope the lab will do this eventually for completion!), the text is overly dismissive of smad4b contribution, i.e. stating extremely low expression (p.6) as example for qualitative statements that should be revisited. The authors provide no evidence for Smad4b protein levels for instance (might the mRNA be more efficiently translated than 4a, for instance?), so all claims as to Smad4b's lack of contribution should be put into proper perspective. The morpholino experiments are indicative, but

hardly definitive evidence compared to the authors' thorough assessment of smad4a. Similarly the statement on lack of compensation based on mRNA, as no protein-level evidence is provided. Revisiting the smad4b-related statements will put this all into context and help the manuscript to gain further traction.

2) The intriguing observation of greatly reduced pSmad1/5 levels upon Smad4a loss is a valuable new insight, yet as presented in its current form it raises more questions than answers. The authors convincingly document the striking reduction of pSmad1/5 and provide evidence that adding BMP ligands to the system restores the levels. Yet, the explanations or mechanistic links of the provided experiments fall short of establishing any proper causal connection.

First, the authors postulate that pSmad1/5 cannot bind their target genes in the absence of Smad4a, and move forward test human SMAD1/5 activity upon reduced SMAD4 for a single target gene locus. This experiment, while providing an interesting example, is hardly representative of the in vivo biology the authors are establishing in the zebrafish embryo and should be dampened in how the result is representative of the overall smad4a mutant phenotype in zebrafish embryos, i.e. without showing reduced Smad1/5 binding chromatin-wide in smad4a mutants the generalization of this finding in a different model is hardly generalizable.

Second, if BMP ligands are not affected by Smad4a loss, what does then cause the strong reduction of pSmad1/5? The authors are encouraged to discuss this point, i.e. might protein stability of Smad1/5 be affected without Smad4a?

Lastly, the authors are encouraged to show the "data not shown" of the lack of BMP4-mediated MZsmad4a rescue.

3) The part on the lack of severe impact on Nodal signaling is a highlight of the manuscript. The authors look at several markers for early mesendoderm and endoderm proper that will be instrumental data points for the field. reduced but not gone.

How do the authors consolidate that there still are residual sox32 cells in the absence of Nodal signaling relay, i.e. what minor redundancy might there be that leads to small amounts of sox32 being expressed? Have the authors looked at any of the more specialized BMP/Nodal targets such as the ventral mesendoderm markers vent and vox, which could provide a more fine-grained view of the observed BMP vs Nodal impairment?

Minor Points:

1) The header "Loss of Smad4a steers embryo development towards a BMP zero morphotype" is brilliant.

In this section, the authors introduce the use of their OPT pipeline, which really is a more quantified approach to classify what has commonly been called "monsters" in the field, i.e. different classes of dorso-ventral patterning issues. Have the authors tested any other developmental insults that lead to morphological changes as quantified in their manuscript? From the description of the results, it seems that making a general call about similar mechanistic causes for the observed deformed embryos is only possible within a given genetic or phenotype space, i.e. within BMP/Nodal perturbations. Might be good to discuss for interested readers that seek to implement this pipeline for their work.

2) Generally, could it be that the smad4a LoF causes amorph or neomorph phenotypes by forcing other Smad complexes that normally wouldn't happen endogenously?

Rebuttal to referees' comments

Referee #1:

Through a sophisticated combination of careful smad4a (and 4b, which does not need further attention and is also not a compensator of smad4a deficiency) gene expression studies in zebrafish early embryos (main figures) and larvae (suppl. figs), CRISPR/Cas based knockout, transcriptomics, live imaging including assessing embryo shape, and also using light sheet microscopy and OPT (the latter having innovative merit on its own, also because of the incorporated morphometry), Gulielmi and co-workers of the team of Caroline Hill present a compelling case for a distinct Smad4a effector dependency for BMP and Nodal signaling and proper regulation of the respective target genes in the developing embryo. It leads to a number of important observations and conclusions of developmental signaling by TGFbeta family subgroup ligands. It is also going to be relevant for congenital and chronic disease, including cancer. Indeed, this work is entirely novel and its results are extremely important for the TGFbeta family signaling field, for it is a significant step beyond previous and more superficial, predominantly cell culture work or studies in worm, fly and vertebrate embryos. This is altogether a superb study about which only minor remarks apply. It will fit perfectly the top-quality requirements by your journal.

We thank reviewer for appreciating both the conceptual and technical aspects of our study and recognising its significance across different domains of biology and disease.

Minor remarks:

The authors are perfectly aware that certainly and par excellence BMP, but also Nodal signaling in the embryo is controlled by autoregulation, synexpression and feedback mechanisms, including on shared or separate components of these two systems, many of which they investigate, including by RNA-seq of MZsmad4a embryos at 50% epiboly (the appropriately selected stage). The already impressive results of this approach could however be discussed more in detail. Assuming they have the RNA-seq results and the data are there already, it would for example be interesting to discuss on top of the downregulated BMP targets within the batch of changed genes (i) also the upregulated genes in the KOs (any of these also known – and biochemically of functionally proven - as modulators of BMP/Nodal signaling?), and report whether (ii) levels of TIF1gamma/Trim33 change (the ratio of this repressor or alternative transcription factor that promotes TGFbeta signaling to Smad4 is important, in particular for short stimulation times that mediate higher threshold responses) and (iii) levels of TTF-1/Nkx2.1 might have changed. Although in particular the latter, for reasons of temporal and domain expression, may be unlikely, it is recommended to have a look at this and perhaps even mention explicitly, as this is a Smad4 complex disruptor (albeit best documented in Smad3-dependent EMT) prohibiting Smad4 binding to chromatin and hence is part of Smad4-independent regulation.

We agree with the reviewer that especially within the dynamic signalling landscape of the developing embryo, the consequences of any signalling manipulation must be

contextualized given the positive and negative feedbacks in the system. While we find such a mechanism operating to control Nodal signalling levels in MZ*smad4a* embryos through *lft1* and *lft2*, we did not find evidence of specific regulators for BMP and Nodal to be upregulated in our RNA-seq dataset. Among the upregulated genes were neural plate markers, as would be expected as their expression is inhibited by BMP signalling (**Fig. S6**). We have now also explored the expression of the target genes suggested by the reviewer, which are indeed important modulators of TGF- β pathway signalling. For the transcription factor TIF1 γ /Trim33, we found that expression was unchanged between WT and MZ*smad4a* mutant at 50% epiboly (Trim33: 0.16 Log₂FC, see **Supplementary file 1**). For the repressor TTF-1/Nkx2.1, we found that, as correctly anticipated by the reviewer, this gene is not expressed in the zebrafish embryo at 50% epiboly.

- In the section addressing the requirement of smad4 for P-Smad1/5 to bind to DNA, reference (i.e. refs. 55-57) is made to binding motifs. However, here it would be appropriate to also cite the identification of CG-rich sequences by Smad1/5 by e.g. the Niehrs group, but certainly also of Morikawa and Miyazono teams (who used BMP-Smad ChIP-seq). The latter teams identified the "GC" motif (high-affinity, so low BMP doses are sufficient) and "GA" motif (low-affinity, so high BMP doses needed) ideally separated by 5 bp from a Smad4 binding site. In this respect, and even taking into account the used dose of 20 ng BMP4/ml – which in the BMP field is an intermediate dose - in the HaCat cells, the analysis of ID3 is relevant, but likely only addresses a "GC" type BMP-Smad/Smad4 motif. The inclusion of a typical "GA" motif containing proximal promoter of another BMP target gene would be recommended here. Also in this system, and similar to one of the aforementioned minor remarks, levels of in particular TIF1 γ /Trim33 could be quickly looked at (and perhaps mentioned) in Smad4 WT versus Smad4 KO cell lines.

We have now added a reference **on page 8** to the Niehrs lab paper, Karaulanov et al 2004 (PMID: 14963489) and also agree with the reviewer that the diversity in pSMAD1/5 binding sites reported in (Morikawa et al., 2012, PMID: 21764776) should be referenced. This has now been added to the text – **see page 8**. However, given that pSMAD1/5 cannot bind to high affinity "GC" motifs in the absence of SMAD4, we think it unlikely that it would bind to low affinity "GA" motifs, the latter being more likely to be dependent on SMAD4. Nevertheless, we have used our RNA-seq dataset in MZ*smad4a* embryos and looked at the expression of *BMPR2* and *JAG1* orthologs, which were classified as low affinity targets by Morikawa and co-workers. We found that while expression of *bmpr2a* and *bmpr2b* was unaffected by the loss of Smad4a, as their transcripts are of maternal origin, expression of *jag1a* and *jag1b* was downregulated in MZ*smad4a* mutants (*jag1a*: -0.37 log₂FC, *jag1b*: -1.52 log₂FC, see **Supplementary file 1**), consistent with the view that pSmad1/5 cannot bind the 'GA' motifs in the absence of Smad4. With respect to the issue of the levels of TIF1 γ , as mentioned above, its levels do not change between the wild type and MZ*smad4a* mutant. We have therefore not pursued this further.

- With the experiments that compellingly show that Smad4-independent Nodal signaling is less robust, and where it is speculated that P-Smad2 homotrimers are recruited by transcription factors, it is along similar lines also recommended to look

at the level of TTF-1 and, if there, to consider carrying out an extra experiment including TTF-1 morpholinos on top of the other set-ups.

We thank the reviewer for the suggestion. However, as TTF-1 is not expressed during the stages examined in the paper, we have not explored this further.

Referee #2:

As requested, my review focuses on the technical aspects of the scRNA-seq analysis included in the manuscript. These analyses were carried out using a pooled sample from 25 embryos. This approach was necessitated by the small numbers of cells that can be obtained from a single embryo and the technical limitations of the scRNA-seq platforms (i.e. fixed scale and limited capture efficiency). However, it can be expected that the amount of inter-individual variability is relatively low during these early stages of development, and therefore the pooling approach does not represent a significant compromise for data interpretation. There appears to be no technical replicates, but as the focus was on analyzing composition of a single sample type (instead of comparisons between samples), in my opinion this is acceptable. The library prep was carried out using fixed cells, adhering to manufacturer's protocols (10x Genomics). The methods section does not contain all the details that might be relevant to readers, such as read lengths, sequencing depth, number of cells loaded and numbers of genes detected per cell.

Low quality cells were filtered out by setting a threshold value for fraction of reads mapping to mitochondrially-encoded genes. This is a straightforward and commonly used approach, although it is often used in combination with other criteria (e.g. number of UMI or genes detected per cell). The fraction of cells discarded is not specified, although it might be of interest to some readers. The data was analyzed using the Seurat toolkit, which is one of the best established and widely used approaches. The analysis was limited to dimensionality reduction and clustering, representing a straightforward "atlasing" approach. The clustering analysis was performed at relatively low resolution, and the resulting clusters appear well defined (Fig S2A) and correlate with the expression of several known marker genes. Altogether, in my opinion these analyses are technically sound and the data appears to support the conclusions made by the authors. However, I would encourage the authors to add some details to the methods section.

We thank the reviewer for the accurate assessment of our scRNA-seq analysis. As suggested by the reviewer we have now added more information to the method section (see Material and Methods, scRNA-seq analysis, **page 22**)

Referee #3:

In the submitted manuscript, Guglielmi and colleagues tackle a surprisingly unresolved issue in developmental biology: the contribution (or lack thereof) of Smad4 to BMP and Nodal signaling. Smad4 as widely touted "helper Smad" is generally assumed to be an essential co-factor for phosphorylated Smads downstream of the major BMP and Nodal responses in vertebrates - yet, most of this

assumption is extrapolated from few in vivo validations and mostly from how the TGF-beta pathways work in Drosophila. The authors take on this interesting challenge by analyzing the first smad4a mutant in zebrafish, revealing that predominantly BMP signaling is perturbed upon Smad4 loss, while Nodal signaling is still functioning rather well.

The authors introduce their work with an excellent intro section and back up their findings with several classic and new means in an outstanding showcase of technical prowess (and mostly friendly color schemes for color-impaired readers). The findings will be of widespread interest to the field. In its current form, the manuscript will benefit from revisiting the claims and details listed below to strengthen the narrative and context of the presented evidence.

We were very pleased that the reviewer finds our study of widespread interest, and we appreciate the positive comments for the array of techniques used in the study

Major Points:

1) The authors go through several iterations of stating that smad4b is unlikely to be of significance to their work and to early Smad4 function in the zebrafish embryo - yet, fact is that smad4b exists, is expressed in early development, and the authors do not provide any genetic evidence for any lack of contribution. While the authors did not seem to have generated a smad4b mutant (the reviewer does hope the lab will do this eventually for completion!), the text is overly dismissive of smad4b contribution, i.e. stating extremely low expression (p.6) as example for qualitative statements that should be revisited. The authors provide no evidence for Smad4b protein levels for instance (might the mRNA be more efficiently translated than 4a, for instance?), so all claims as to Smad4b's lack of contribution should be put into proper perspective. The morpholino experiments are indicative, but hardly definitive evidence compared to the authors' thorough assessment of smad4a. Similarly, the statement on lack of compensation based on mRNA, as no protein-level evidence is provided. Revisiting the smad4b-related statements will put this all into context and help the manuscript to gain further traction.

We agree with the reviewer that ruling out any Smad4b contribution is important for the study, given that Nodal signalling is still functional in MZ*smad4a* mutants. We also agree with the reviewer that looking at Smad4b protein levels would be optimal. However, we have not been able to find any Smad4 antibodies that recognise either of the zebrafish Smad4s, ruling out any analysis of the protein levels of Smad4a and Smad4b at present. We have, however, undertaken a series of careful experiments to eliminate the possibility of a role for Smad4b. We have shown by extensive qPCR analysis and RNA-seq analysis that levels of *smad4b* mRNA are extremely low during early zebrafish embryo development (**Fig. 1A, Supplementary Fig. 1A, H**). Also, levels of *smad4b* mRNA are insensitive to changes in *smad4a* levels, thus eliminating the possibility of any compensatory upregulation (**Supplementary Fig. 1I**). Given this and the dominant role of Smad4a, we do not think that generation of a *smad4b* mutant would provide further insight into the role of Smad4 in early zebrafish development. Indeed, we have provided qualitative evidence that specific

knockdown of *smad4b* does not affect residual expression of Nodal target genes in the absence of *smad4a* (**Supplementary Fig. 3C, D**). To further address the reviewer's point and strengthen our conclusions, we have now deleted SMAD4 in mouse ESCs, where there is only one *Smad4* gene. Notably, upon BMP4 stimulation, expression of BMP target genes was abolished in SMAD4 KO mESCs (**Supplementary Fig. 4D**). However, consistent with our observations in zebrafish embryos, a subset of early Nodal target genes (*T*, *Lefty1*, *Lefty2*) were robustly induced upon Activin A stimulation (**Supplementary Fig. 4C**). This further demonstrates that Nodal signalling can function without a functional *Smad4* gene in both zebrafish and mouse. See also text on **page 8**.

2) The intriguing observation of greatly reduced pSmad1/5 levels upon Smad4a loss is a valuable new insight, yet as presented in its current form it raises more questions than answers. The authors convincingly document the striking reduction of pSmad1/5 and provide evidence that adding BMP ligands to the system restores the levels. Yet, the explanations or mechanistic links of the provided experiments fall short of establishing any proper causal connection.

First, the authors postulate that pSmad1/5 cannot bind their target genes in the absence of Smad4a, and move forward test human SMAD1/5 activity upon reduced SMAD4 for a single target gene locus. This experiment, while providing an interesting example, is hardly representative of the in vivo biology the authors are establishing in the zebrafish embryo and should be dampened in how the result is representative of the overall smad4a mutant phenotype in zebrafish embryos, i.e. without showing reduced Smad1/5 binding chromatin-wide in smad4a mutants the generalization of this finding in a different model is hardly generalizable.

We thank the reviewer for the suggestions. However, we are confident that our mammalian tissue system provides a representative model for testing pSMAD1/5 binding to DNA in the absence of SMAD4. Indeed, the SMAD1/5–SMAD4 binding sites at the *ID3* locus used in our assay, are conserved between human, mouse and zebrafish (Hill lab unpublished data). The choice of a mammalian tissue model was driven by the higher tractability of the system, especially for biochemical assays relying on nuclear extracts.

Concerning the greatly reduced pSmad1/5 levels in MZ*smad4a* embryos - as the reviewer points out, we have shown that pSmad1/5 levels can be restored by ectopic expression of BMP ligands, which demonstrates that the loss of pSmad1/5 is due to the lack of zygotic expression of Bmp ligands. We presume these to be *Bmp2b* and *Bmp7*, as zygotic expression of these ligands requires intact Bmp signalling downstream of *Gdf6a*, as mentioned on page 9. We now show that injection of *BMP4* mRNA does not rescue the dorsalized phenotype in MZ*smad4a* (**Supplementary Fig. 5G, H**), proving that in zebrafish, *Smad4a* is required downstream of Bmp ligands as well as upstream. Furthermore, we have now broadened our study to include mouse ESCs. Consistent with the view that SMAD4 is required for binding of pSMAD1/5 to chromatin, we now show that in SMAD4 KO mESCs, BMP stimulation induces strong activation of pSmad1/5, but expression of BMP target genes is totally abolished (**Supplementary Fig. 4A and D**).

Second, if BMP ligands are not affected by Smad4a loss, what does then cause the strong reduction of pSmad1/5? The authors are encouraged to discuss this point, i.e. might protein stability of Smad1/5 be affected without Smad4a? Lastly, the authors are encouraged to show the "data not shown" of the lack of BMP4-mediated MZsmad4a rescue.

In our RNAseq analysis, the BMP ligand *bmp4* is strongly downregulated in MZ*smad4a* mutant. However, as the reviewer points out, expression of, in particular, *bmp2b*, does not appear to be downregulated in MZ*smad4a* mutants. We think that this is because the earliest expression of *bmp7* and *bmp2b* is not under the control of the Bmp pathway but is regulated by the maternal POU domain transcription factor Pou2/Pou5f1. There may also be a maternal contribution to the expression of *bmp7* and *bmp2b*. However, as outlined in the Discussion (**page 15/16**), these early *bmp2b/7a* transcripts are either not efficiently translated, or abundant enough to efficiently promote strong initiation and propagation of the BMP signalling gradient. This would be consistent with the absence of a phenotype in maternal *bmp7a* mutants. To further explore this issue, we have analysed an RNAseq dataset for zygotic *bmp7*^{-/-} mutants at 8 hpf. We find that the MZ*smad4a* mutant are transcriptionally equivalent, with respect to BMP target genes to embryos lacking all zygotic *bmp7* (**Supplementary Fig. S3B and S6A and text on pages 8 and 12**). Finally, as suggested by the reviewer we have now added our "data not shown" for the lack of a BMP4-mediated MZ*smad4a* rescue in **Supplementary Fig. 5G** and it is quantified in **Supplementary Fig. 5H**.

3) The part on the lack of severe impact on Nodal signaling is a highlight of the manuscript. The authors look at several markers for early mesendoderm and endoderm proper that will be instrumental data points for the field. reduced but not gone. How do the authors consolidate that there still are residual sox32 cells in the absence of Nodal signaling relay, i.e. what minor redundancy might there be that leads to small amounts of sox32 being expressed? Have the authors looked at any of the more specialized BMP/Nodal targets such as the ventral mesendoderm markers vent and vox, which could provide a more fine-grained view of the observed BMP vs Nodal impairment?

We thank the reviewer for raising a point that was perhaps not sufficiently clear in the manuscript. We have shown that even though less efficient, the Nodal signalling relay is still functional in MZ*smad4a* mutants, at both the level of pSmad2 activity and target gene expression. Therefore, residual expression of the endoderm markers *sox32* and *sox17* in MZ*smad4a* mutants is a direct consequence of Nodal signalling activity. Consistent with these data, *mixl1*, which is required downstream Nodal signalling to induce endoderm specification (van Boxtel et al. 2018), is still expressed in MZ*smad4a* mutants (**Fig. 2B**). Following the reviewer's suggestion, we have looked at expression of *vent* and *vox* in MZ*smad4a* mutants (*vent*: -0.51 log2FC, *vox*: -0.23 log2FC) which were mildly downregulated. However, expression of these genes depends on additional signalling inputs beside Nodal and BMP, like Wnt signalling (Ramel and Lekven, 2004) and therefore may not be informative in the specific context of our study. Also, these genes are typically expressed at the

ventral side of the embryo. Of note, we show that *sox32*-positive cells are specified also on the dorsal side of *MZsmad4a* embryos (**Fig. 5C**).

Minor Points:

1) The header "Loss of Smad4a steers embryo development towards a BMP zero morphotype" is brilliant. In this section, the authors introduce the use of their OPT pipeline, which really is a more quantified approach to classify what has commonly been called "monsters" in the field, i.e. different classes of dorso-ventral patterning issues. Have the authors tested any other developmental insults that lead to morphological changes as quantified in their manuscript? From the description of the results, it seems that making a general call about similar mechanistic causes for the observed deformed embryos is only possible within a given genetic or phenotype space, i.e. within BMP/Nodal perturbations. Might be good to discuss for interested readers that seek to implement this pipeline for their work.

We thank the reviewer for the positive comments, and we are pleased that the reviewer appreciates our morphometric analysis of *MZsmad4a* embryos. So far, we have looked at morphological changes upon BMP (**Fig. 7**) and Nodal signalling perturbations (**Supplementary Fig. 8**). In the future we are planning to test additional perturbations of other signalling pathways. As the reviewer rightly points out, characterization of mutant morphology cannot be determined solely by looking at changes in morphological parameters. It must be contextualized within defined morphological spaces. We have now mentioned this in the Discussion (see **page 18**).

2) Generally, could it be that the smad4a LoF causes amorph or neomorph phenotypes by forcing other Smad complexes that normally wouldn't happen endogenously?

We thank the reviewer for making a very interesting point. Yes, loss of Smad4a can force new signalling complexes, as homomeric pSmad2 and homomeric pSmad1/5 complexes do accumulate in the nucleus in the absence of Smad4a. In the case of pSmad2 they can also induce gene expression, likely recruited by Foxh1 (see Fig. 5A). However, from a phenotypic point of view, we find that the main morphogenetic defects in *MZsmad4a* are rather independent of any complex reconfiguration, as they are primarily caused by the complete loss of BMP-dependent transcription.

Reviewers' Comments:

Reviewer #1:

Remarks to the Author:

In the revision of their already excellent manuscript (see initial review report) the authors have adequately dealt with each of the comments and suggestions this reviewer had, so it can be accepted as is.

Reviewer #3:

Remarks to the Author:

Upon the previous submission of the manuscript, I raised minor concerns regarding lack of details in the methods section. In the revised manuscript these criticisms have been diligently addressed and I do not have any further concerns.

Reviewer #4:

Remarks to the Author:

The authors have addressed all raised points to the reviewer's satisfaction within the framework of feasible experiments at this time. Congratulations on an insightful study that contributes significant new details to the field of early embryo patterning.

Rebuttal to referees' comments

Reviewer #1 (Remarks to the Author):

In the revision of their already excellent manuscript (see initial review report) the authors have adequately dealt with each of the comments and suggestions this reviewer had, so it can be accepted as is.

We thank the reviewer for their very positive assessment of our work and are pleased that the reviewer thinks the paper can be accepted as it is.

Reviewer #3 (Remarks to the Author):

Upon the previous submission of the manuscript, I raised minor concerns regarding lack of details in the methods section. In the revised manuscript these criticisms have been diligently addressed and I do not have any further concerns.

We are pleased that the reviewer was satisfied with our revisions.

Reviewer #4 (Remarks to the Author):

The authors have addressed all raised points to the reviewer's satisfaction within the framework of feasible experiments at this time. Congratulations on an insightful study that contributes significant new details to the field of early embryo patterning.

We thank the reviewer for their very positive comments on our paper.